# EgoExoBench: A Benchmark for First- and Third-person View Video Understanding in MLLMs

**Yuping He**[1,2*], **Yifei Huang**[2,3*], **Guo Chen**[1], **Baoqi Pei**[2,4], **Jilan Xu**[2,5]
**Tong Lu**[1†] and **Jiangmiao Pang**[2†]
[1]Nanjing University, [2]Shanghai AI Laboratory, [3]The University of Tokyo
[4]Zhejiang University, [5]Fudan University
https://github.com/ayiyayi/EgoExoBench

## Abstract

Transferring and integrating knowledge across first-person (egocentric) and third-person (exocentric) viewpoints is intrinsic to human intelligence, enabling humans to learn from others and convey insights from their own experiences. Despite rapid progress in multimodal large language models (MLLMs), their ability to perform such cross-view reasoning remains unexplored. To address this, we introduce EgoExoBench, the first benchmark for egocentric-exocentric video understanding and reasoning. Built from publicly available datasets, EgoExoBench comprises over 7,300 question–answer pairs spanning eleven sub-tasks organized into three core challenges: semantic alignment, viewpoint association, and temporal reasoning. We evaluate 13 state-of-the-art MLLMs and find that while these models excel on single-view tasks, they struggle to align semantics across perspectives, accurately associate views, and infer temporal dynamics in the ego-exo context. We hope EgoExoBench can serve as a valuable resource for research on embodied agents and intelligent assistants seeking human-like cross-view intelligence.

## 1   Introduction

Understanding and transferring knowledge between first-person (egocentric) and third-person (exocentric) perspectives is a fundamental aspect of human intelligence [17, 27]. We routinely learn new skills by observing third-person demonstrations and mapping those actions onto our own egocentric experience [3, 20, 53, 24, 21]. Conversely, we can adopt an exocentric view to articulate insights and guide others [9, 5]. Beyond simple transfer, humans excel at cross-view joint reasoning: in procedural tasks, we align live egocentric observations with exocentric examples to anticipate next steps [27, 28]; in navigation, we fuse egocentric experience with a global map to plan efficient routes [6, 13, 33]. This dual-view reasoning enables robust perception and skill acquisition in dynamic environments.

Replicating this ego–exo reasoning in artificial systems offers a promising path toward more capable embodied agents. Robots that could align large-scale exocentric knowledge, such as instructional video, can execute tasks more effectively and better perform imitation learning [58, 1, 57]. Bridging first- and third-person perspectives is also essential for seamless human–robot collaboration [41, 66].

Recent benchmarks [45, 14, 84, 38, 8] have driven significant advances in single-view video understanding for multimodal large language models (MLLMs) [85, 62, 81, 35, 78]. For example, EgoSchema [45] evaluates temporal reasoning in egocentric streams, Video-MME [14] evaluates multi-modal QA on diverse third-person footage, and MLVU [84] tests long-form video comprehension. While each benchmark pushes the boundary of MLLMs in video understanding from different perspectives, they all operate on either egocentric or exocentric data in isolation. There is

---

[*]equal contribution. [†] corresponding authors

no systematic evaluation of a model's ability to integrate information across ego–exo viewpoints, leaving the question of whether MLLMs can, like humans, perform cross-view reasoning.

To fill this gap, we introduce EgoExoBench, the first benchmark designed specifically for cross-view video understanding in MLLMs. EgoExoBench is built from publicly available paired egocentric–exocentric video sources and curated through a rigorous annotation protocol. It comprises over 7,300 question–answer pairs that target three key dimensions: Ego-Exo Matching, Ego-Exo View Transition, and Ego-Exo Temporal Reasoning. Ego-Exo Matching evaluates semantic alignment by testing whether models recognize the same entities or actions across first- and third-person views. Ego-Exo View Transition probes spatial correspondence, measuring a model's ability to translate between egocentric and exocentric viewpoints. Ego-Exo Temporal Reasoning examines temporal sequence reasoning by asking models to align and predict event sequences across asynchronous or overlapping video streams. 11 subtasks span these three dimensions, forming a comprehensive evaluation suite for ego–exo view understanding and reasoning.

We conduct extensive experiments on 13 open- and closed-source MLLMs, including the GPT-4o [29], GPT-o4-mini [47], Claude 3.7 Sonnet [2], Qwen2.5-VL [62], and InternVL3 [85]. The results reveal a clear pattern: models that perform strongly on single-view tasks experience a significant drop when confronted with cross-view challenges. Further analysis indicates that even models equipped with explicit reasoning capabilities struggle to interleave text-based reasoning with information from dual-view videos. These results highlight a significant gap between current capabilities and the human-like cross-view understanding required for embodied agents and human–robot collaboration. We hope EgoExoBench will serve as a comprehensive evaluation suite for ego–exo view reasoning and inspire new architectures and training strategies to bridge this gap.

## 2 Related Work

**Video benchmarks.** With the rapid application of Multimodal Large Language Models (MLLMs) into video understanding [37, 81, 48], many efforts have been made to benchmark the video understanding capabilities of MLLMs. Benchmarks such as Next-QA [69], Next-GQA [70], MLVU [84], LV-Bench [64], Video-MME [14], and CG-Bench [8] evaluate multiple aspects from general QA to multimodal understanding. Egocentric datasets [10, 16, 42, 51, 4, 11, 31, 80, 60] spur EgoSchema [45] to probe egocentric long-form video understanding, while HourVideo [7] tests spatial intelligence. However, these efforts each focus on a single viewpoint, either egocentric or exocentric, and none assess a model's ability to integrate both. EgoExoBench fills this gap by offering the first benchmark for cross-perspective video understanding, measuring semantic, spatial, and temporal reasoning between first- and third-person streams.

**Ego-exo datasets.** Prior work has produced a variety of paired egocentric–exocentric video collections. Charades-Ego [59] and Home Action Genome [52] capture synchronized first- and third-person video in home environments, while Assembly101 [56] and LEMMA [30] offer both synchronous recordings of multi-step tasks. Collections like Ego-Exo4D [17] and EgoExoLearn [27] offer extensive scale and rich annotations, but lack any downstream QA or reasoning benchmarks. Robotics-focused datasets like iGibson [36] and H2O [34] record fixed-camera arrays for pose estimation and navigation, yet remain narrowly scoped to those tasks. Built atop these datasets, EgoExoBench provides a unified testbed for evaluating cross-view video understanding and reasoning of MLLMs.

**Ego-exo video understanding.** Egocentric video understanding methods often leverage larger-scale exocentric data [46, 18, 61] to compensate for the limited size of first-person corpora. Prior work falls into three main categories: joint view-invariant learning [75, 63, 72, 71, 23, 25], domain adaptation [68, 79, 73, 26], and knowledge distillation [39, 50, 22]. While these approaches improve recognition performance, they mainly focus on recognition problems and do not evaluate a model's ability to reason across perspectives, which is the aspect EgoExoBench is specifically designed to benchmark.

## 3 Benchmark

### 3.1 Task Suite

EgoExoBench is a large-scale benchmark for evaluating cross-view video understanding in multimodal large language models. It covers diverse environments and activities, probing three key

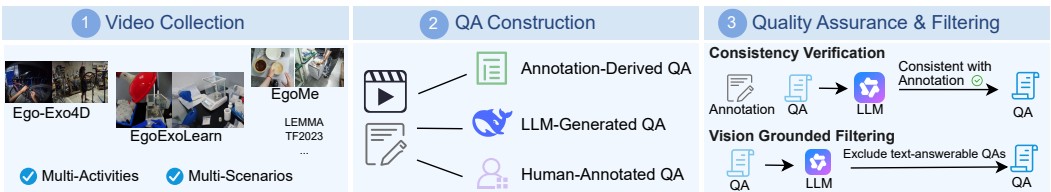

Figure 1: The Construction Pipeline of EgoExoBench.

dimensions of ego–exo reasoning: semantic alignment, spatial correspondence, and temporal reasoning. While open-ended question answering closely mirrors human dialogue, automating the evaluation of free-form responses can be difficult and error-prone. To enable reliable, scalable assessment of cross-view video understanding, EgoExoBench adopts a multiple-choice question (MCQ) format. In the following sections, we detail our task suite and question–answer generation pipeline, both crafted to produce diverse, high-quality multiple-choice questions that rigorously probe semantic alignment, spatial correspondence, and temporal reasoning.

Creating a benchmark for cross-view video understanding presents unique challenges: questions must not only span multiple temporal segments but also require synthesizing information between first- and third-person perspectives. EgoExoBench addresses this by first establishing three core dimensions that capture the essence of ego–exo reasoning: semantic alignment, spatial correspondence, and temporal relation. We then leverage rich, publicly available egocentric–exocentric datasets, mining their synchronized and asynchronous multi-view recordings, spatial annotations (e.g., poses, bounding boxes), and temporal action labels to construct high-quality MCQs.

Our task suite comprises 11 subtasks organized under 3 key dimensions. Ego–Exo Matching evaluates semantic alignment across perspectives; Ego–Exo View Transition measures spatial correspondence between egocentric and exocentric coordinate frames; and Ego–Exo Temporal Reasoning examines sequence integration by requiring models to align and predict event flows across paired video streams. Together, these subtasks form a comprehensive evaluation suite for ego–exo view video understanding and reasoning.

### 3.2 Data Construction Pipeline

As shown in Figure 1, our QA creation follows a three-stage pipeline tailored to the specific demands of cross-view evaluation. Below, we describe the general construction process. We provide task-specific details in subsequent sections.

**Video Collection.** Our benchmark aggregates videos from six ego-exo datasets: Ego-Exo4D [17], EgoExoLearn [27], LEMMA [30], EgoMe [49], TF2023, and CVMHAT [19]. These datasets span a wide range of environments (*e.g.*, kitchen, laboratory, sports field) and activities (*e.g.*, cooking, sports, repair). Ego-Exo4D [17], LEMMA [30], TF2023 [82], and CVMHAT [19] include synchronized multi-view videos, while EgoExoLearn [27] and EgoMe [49] feature asynchronous demonstration-follower recordings. This diverse setting allows for a comprehensive evaluation of models' ability to relate and understand both synchronized and asynchronous cross-view scenarios.

**Question-Answer Construction.** To accommodate the diverse nature of tasks in our benchmark, we employ multiple QA construction strategies. *(1) Annotation-Derived QA.* For tasks with structured and deterministic annotations, we construct QA pairs directly from the annotations using predefined templates. *(2) LLM-Generated QA.* For cases that require open-ended reasoning in QA generation, we utilize LLMs [40, 76] to generate context-aware QA pairs. The LLMs are provided with task definitions, detailed annotations, exemplar QA pairs, and task-specific constraints. *(3) Human-Annotated QA.* For tasks that require fine-grained spatial understanding, current MLLMs fail to produce reliable results. Therefore, we employ human annotators to construct the corresponding QA pairs. All QA pairs are formatted as 4-way multiple-choice questions.

**Quality Assurance and Filtering.** To ensure the reliability of the constructed QA pairs, we implement a quality assurance and filtering process. *(1) Consistency Verification.* We perform an automated consistency check by prompting an LLM [76] to verify the logical correctness of each question-answer pair against the original video annotations. Questions are discarded if they are

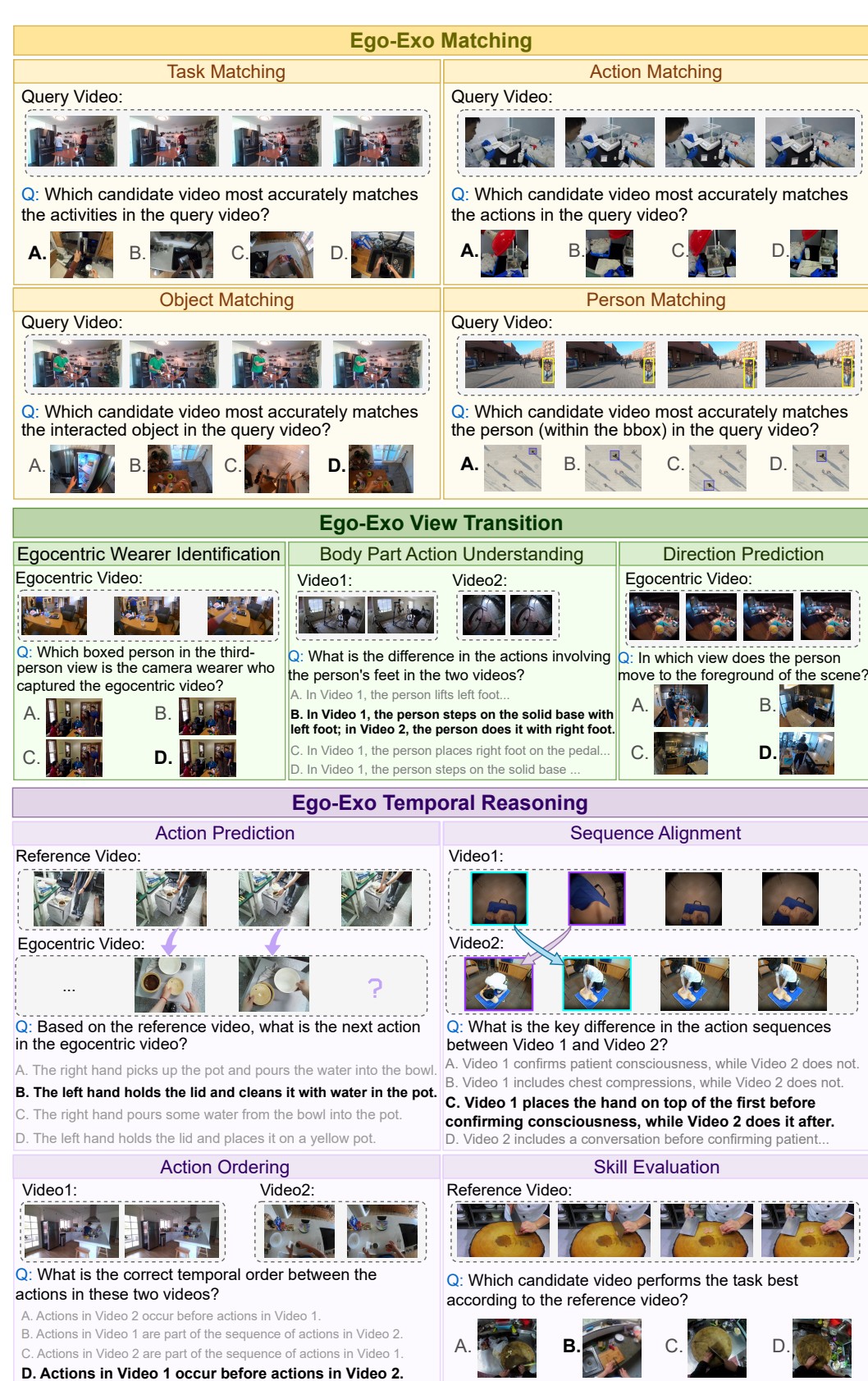

Figure 2: Example MCQs in EgoExoBench. The correct answers are bold-highlighted.

ambiguous, inconsistent with annotations, or admit multiple plausible answers. *(2) Vision-Grounded Filtering.* To guarantee that each question necessitates visual understanding, we present text-only questions to an LLM [76] and discard QA pairs that can be answered solely based on textual input.

## 3.3 Task-Specific Details

### 3.3.1 Ego-Exo Matching

The Ego–Exo Matching dimension evaluates a model's ability to semantically align content across first- and third-person views, which is a critical foundation for any cross-view reasoning. We design four subtasks: task-, action-, person-, and object-level matching, aiming for a comprehensive assessment.

**Task Matching (TM).** We draw on synchronized and asynchronous demonstrations in Ego-Exo4D [17], EgoExoLearn [27], and LEMMA [30]. We pair egocentric clips of a high-level activity (*e.g.*, assembling a device in a kitchen) with exocentric videos labeled with the same task ID. To prevent trivial scene-matching, ground-truth pairs come from different environments within the same scenario, while negative candidates depict other tasks in visually similar contexts. QA pairs are generated via annotation-derived templates, converting task IDs into four-choice MCQs, and are filtered through automated consistency checks and vision-grounded prompts to ensure reliance on visual input.

**Action Matching (AM).** Given a video clip, this task aims to identify a corresponding clip from another viewpoint that captures the same action. It requires bridging perspective-induced variations, such as camera motion and occlusion, to establish fine-grained action correspondence. We construct QA pairs using videos from LEMMA [30] and EgoExoLearn [27]. For LEMMA [30], the ground-truth clip is temporally aligned with the query but recorded from a different viewpoint. For EgoExoLearn [27], the correct matching clip is drawn from a different video depicting the same action.

**Object Matching (OM).** Given a query video, this task aims to identify the candidate video from another viewpoint that involves interaction with the same object. We curate videos from LEMMA [30] and derive object interaction from the annotations. To increase difficulty, no textual cues about the object are given. The model must first infer the object interacted with in each video before matching across views. The correct video is temporally aligned with the query but from another viewpoint, while negative candidates depict interactions with different objects in the same environment.

**Person Matching (PM).** Given a video clip, this task aims to identify the same individual from a different viewpoint. We curate data from CVMHAT [19], which provides synchronized egocentric and top-view recordings in outdoor scenarios. In each query, the target individual is highlighted using a bounding box. The correct answer corresponds to the same person observed at the same timestamp but from another viewpoint. To prevent shortcuts based on scene or temporal cues, negative candidates are drawn from the same timestamp but depict different individuals.

### 3.3.2 Ego-Exo View Transition

The subtasks in Ego–Exo View Transition assess a model's ability to translate spatial information between first- and third-person perspectives. We define three subtasks: egocentric wearer identification, direction prediction, and body part action understanding, each constructed with tailored data sources and QA strategies.

**Egocentric Wearer Identification (EWI).** Given an egocentric video, this task aims to identify the camera wearer in a third-person view. Compared to the Person Matching task, this setting is more challenging, as the egocentric wearer is typically absent from their own viewpoint. With limited appearance cues, the model must infer spatial relationships by analyzing the relative positions of surrounding people and objects, and then map these relations from the egocentric to the exocentric perspective. We build upon the TF2023 dataset [82], which provides synchronized egocentric–exocentric image pairs and annotated human bounding boxes. We retain only samples containing four or more individuals to increase spatial complexity. To construct MCQ candidates, we select a different person based on the same exocentric frame.

**Direction Prediction (DP).** Given an egocentric video depicting the motion of a person or an interacted object, the task is to identify, among four provided third-person views, which view shows

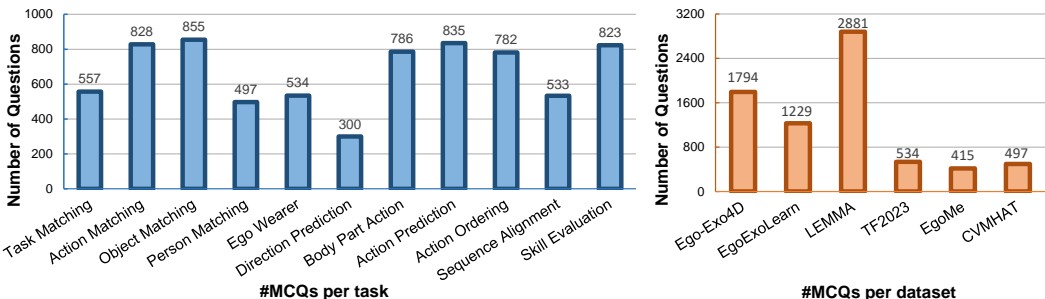

Figure 3: Distribution of MCQs in EgoExoBench. Left: number of questions per subtask. Right: number of questions per dataset.

the motion in a specified direction (e.g., left to right). The direction is specified in the textual question. Using Ego-Exo4D's [17] synchronized multi-view recordings, we first identify segments with clear entity movements via Qwen2.5 32B-filtered narrations. Expert annotators then label the movement direction of the wearer or an object in the egocentric view. Each question offers multiple exocentric clips, only one of which matches the specified directional condition. Manual verification ensure that only unambiguous, visually grounded examples remain.

**Body Part Action Understanding (BPA).** This task probes fine-grained spatial mapping of limb movements across the two views. We extract segments from Ego-Exo4D [17] and EgoExoLearn [27] where narrations explicitly reference body parts such as the left or right hand. Qwen2.5-32B [76] identifies associated verbs and objects, and DeepSeek-V3 [40] generates distractors that share at least one action or object cue. We generate questions asking to pair egocentric and exocentric clips showing the same body-part activity. Finally, we apply a Qwen2.5-32B-based filter that discards any instances solvable via text alone.

### 3.3.3 Ego-Exo Temporal Reasoning

The Ego–Exo Temporal Reasoning evaluates a model's ability to align and infer the flow of events across egocentric and exocentric video streams. We define four subtasks—next-action prediction, action ordering, sequence alignment, and skill evaluation.

**Action Prediction (AP).** We pair a partial egocentric clip with a longer exocentric demonstration and ask models to forecast the subsequent egocentric action. To construct data, we collect videos from LEMMA [30] and EgoMe [49]. For LEMMA [30], we utilize the action annotations to extract ground-truth next actions and negative candidates. Specifically, we select ten actions surrounding the current action in the egocentric sequence, excluding the true next action, and prompt Qwen2.5-32B [76] to identify the three most plausible distractors from this set. In asynchronous settings from EgoMe [49], we rely on annotated action descriptions from the egocentric view and use DeepSeek-V3 [40] to generate three distractors accordingly. Finally, we use Qwen2.5-32B [76] to filter out questions that can be answered based solely on textual input.

**Action Ordering (AO).** In this task, the model is given two short clips, one egocentric, one exocentric, and is required to judge their temporal relationship. To construct data, we begin with raw action annotations from LEMMA [30]. Each question instance is formed by selecting two temporally continuous action clips. To ensure the validity of temporal ordering, we prompt Qwen2.5-32B [76] with textual descriptions of the actions to filter out ambiguous or unordered action pairs.

**Sequence Alignment (SA).** Sequence alignment extends ordering to multi-step activities by asking whether an egocentric and an exocentric video share the same action ordering, are reversed, or differ at key steps. To construct the data, we first curate videos from Ego-Exo4D [17] that include keystep annotations. Video pairs are selected from the same activity category, with each pair containing at least two shared steps. For each pair, we organize the step descriptions and use DeepSeek-V3 [40] to generate questions that emphasize key differences in action sequences. To ensure question quality, we employ Qwen2.5-32B [76] to verify that the correct answer aligns with the annotated steps. Finally, we used Qwen2.5-32B [76] to filter out questions that could be answered with only textual input.

**Skill Evaluation (SE).** Given an expert demonstration video as reference, this task assesses the skill level of other videos from different viewpoints. It is formulated as a multiple-choice question where the model selects the best or worst performance among candidates. Compared to textual guidelines, expert demonstrations offer fine-grained behavioral cues that are often missed by rule-based methods. We construct data from EgoExoLearn [27] and Ego-Exo4D [17]. For EgoExoLearn [27], we use exocentric demonstration videos as references and build transitive ranking chains from pairwise skill annotations to establish a clear hierarchy among candidate videos. For Ego-Exo4D [17], we select "Late Expert" videos as references and curate candidates with varying proficiency scores from the same activity. Egocentric videos with poor visibility of actions are manually excluded.

### 3.4 Benchmark Statistics

EgoExoBench comprises a total of 7,330 multiple-choice questions (MCQs), each designed in a 4-way format. These questions are constructed from six ego-exo paired datasets, encompassing a wide range of scenarios and activities. As illustrated in Figure 2, the benchmark comprises three task categories encompassing a total of eleven subtasks. Figure 3 presents the distribution of MCQs across subtasks and datasets.

## 4 Experiment

### 4.1 Experiment Setup

We evaluate EgoExoBench in a zero-shot multiple-choice question answering (MCQ) format using a diverse set of MLLMs. Closed-source models include Claude 3.7 Sonnet [2], GPT-4o [29], and GPT-o4-mini [47]. For open-source models, we assess Qwen2.5-VL [62], InternVL3 [85], LLaVA-OV [35], LLaVA-Video [81], NVILA [43], and EgoGPT [78], covering a diverse set of architectures and parameter scales. Each model receives the same standardized prompt, which presents the question stem followed by labeled options and instructs the model to return only the letter of its chosen answer. We apply a rule-based approach to extract the predicted answer. All evaluations use accuracy as the primary metric, and no fine-tuning or gradient updates are performed to ensure fairness. For closed-source models, we conduct evaluations via official APIs. For open-source models, all experiments are performed using four A100 GPUs with 80GB memory each.

### 4.2 Main Result

Prior to the main experiments, we first confirmed that MLLMs reliably distinguish multiple video streams as separate inputs, validating our multi-video experimental setup (Details in supplementary).

Table 1 summarizes the performance of both open-source and closed-source models on EgoExoBench. Among open-source models, Qwen2.5-VL-72B [62] achieves the highest overall accuracy at 47.0%, GPT-o4-mini [47] leads the closed-source group with 48.0%. In the Ego-Exo Matching category, GPT-o4-mini [47] outperforms Qwen2.5-VL-72B [62] by 5% on average, indicating that open-source architectures still lag behind state-of-the-art closed-source systems in explicit semantic alignment across views. For Ego-Exo View Transition and Ego-Exo Temporal Reasoning, Qwen2.5-VL-72B [62] achieves average accuracies of 47.3% and 37.4%, respectively. Its relative strength in spatial and video grounding likely contributes to better performance on these spatial correspondence and sequence integration tasks. Notably, EgoGPT [78], despite being fine-tuned on a large egocentric video corpus (EgoIT-99K) [78], offers only marginal gains over similarly sized counterparts, suggesting that specialized pretraining alone is insufficient to master cross-perspective reasoning without task-specific objectives.

**Human Performance.** To contextualize model performance, we randomly sample 30 questions from each subtask, resulting in 330 questions, and measure human accuracy on the same MCQs. Human evaluators are instructed to derive their answers within 1 minute (quick responses) and 3 minutes (deliberate responses), respectively. As shown in Table 1, humans performing under the deliberate condition (3 minutes per question) achieve an average accuracy of 90.1%, outperforming the best model by 42%. This gap indicates that MLLMs still fall significantly short of human-level in cross-view video understanding. The largest gap appears in the Egocentric Wearer Identification (EWI) subtask, where all human evaluators can successfully infer the identity of the egocentric camera wearer in third-person views based on spatial relationships between people and objects. In contrast,

Table 1: Performance of various open-source and closed-source MLLMs on EgoExoBench. The best result among all models is highlighted in bold, and the second-best is underlined.

| | Avg. | Matching | | | | View Transition | | | Temporal Reasoning | | | |
|---|---|---|---|---|---|---|---|---|---|---|---|---|
| | | TM | AM | OM | PM | EWI | DP | BPA | AP | AO | SA | SE |
| Human Performance (330 Questions) | | | | | | | | | | | | |
| Human (1min) | 64.6 | 72.2 | 58.9 | 62.2 | 70.0 | 91.1 | 78.9 | 53.3 | 68.9 | 61.1 | 60.0 | 34.4 |
| Human (3min) | 90.1 | 95.5 | 90.0 | 92.2 | 94.4 | 100.0 | 96.7 | 85.5 | 91.1 | 87.8 | 88.9 | 68.9 |
| Qwen2.5-VL-72B | 48.5 | 56.7 | 46.7 | 66.7 | 56.7 | 56.7 | 43.3 | 46.7 | 46.7 | 43.3 | 43.3 | 26.7 |
| Claude-3.7-Sonnet | 32.8 | 47.1 | 30.0 | 36.7 | 46.7 | 33.3 | 30.0 | 26.7 | 20.0 | 30.0 | 30.0 | 30.0 |
| GPT-4o | 38.5 | 53.8 | 30.0 | 60.0 | 56.7 | 33.3 | 26.7 | 30.0 | 26.7 | 33.3 | 43.3 | 30.0 |
| Open-source MLLMs | | | | | | | | | | | | |
| Qwen2.5-VL-7B | 32.8 | 40.2 | 34.4 | 45.5 | 36.0 | 26.4 | 30.7 | 34.5 | 19.2 | 31.0 | 37.1 | 26.1 |
| Qwen2.5-VL-32B | 39.7 | 43.3 | 40.1 | 50.4 | 42.3 | 41.7 | 34.0 | 42.4 | 38.6 | 31.1 | 45.8 | 27.5 |
| Qwen2.5-VL-72B | 44.7 | 51.0 | 43.5 | 56.6 | 49.7 | 56.7 | 37.0 | 48.1 | 39.9 | 33.6 | 46.1 | 29.9 |
| InternVL3-8B | 31.3 | 36.8 | 30.9 | 37.0 | 27.2 | 16.5 | 33.7 | 37.2 | 27.4 | 29.8 | 47.2 | 20.9 |
| InternVL3-14B | 35.1 | 38.8 | 31.9 | 43.5 | 31.8 | 30.5 | 29.0 | 41.1 | 33.9 | 32.9 | 48.1 | 24.2 |
| InternVL3-78B | 41.4 | 50.6 | 37.3 | 48.9 | 39.0 | 46.9 | 32.3 | 38.0 | 51.0 | 31.5 | 50.6 | 29.5 |
| LLaVA-OV-7B | 29.5 | 30.7 | 28.7 | 34.0 | 28.0 | 22.7 | 27.0 | 31.4 | 21.7 | 29.3 | 44.1 | 26.8 |
| LLaVA-Video-7B | 31.2 | 33.9 | 29.1 | 35.4 | 27.0 | 29.0 | 27.0 | 36.4 | 23.5 | 28.9 | 43.8 | 28.9 |
| NVILA-8B | 29.6 | 30.0 | 25.0 | 30.8 | 31.6 | 23.6 | 27.0 | 37.7 | 24.4 | 26.2 | 44.5 | 24.8 |
| EgoGPT-7B | 29.6 | 29.8 | 29.5 | 35.6 | 30.4 | 22.3 | 26.3 | 32.1 | 22.3 | 29.4 | 40.1 | 27.6 |
| Closed-source MLLMs | | | | | | | | | | | | |
| Claude-3.7-Sonnet | 31.3 | 33.4 | 33.0 | 35.1 | 30.6 | 37.8 | 30.0 | 33.3 | 26.0 | 28.5 | 30.2 | 26.4 |
| GPT-4o | 38.5 | 52.9 | 44.6 | 57.4 | 48.0 | 41.6 | 24.1 | 37.1 | 27.7 | 26.1 | 37.4 | 26.3 |
| GPT-o4-mini | 48.0 | 65.3 | 45.2 | 56.8 | 53.7 | 73.8 | 30.0 | 40.0 | 46.4 | 35.9 | 51.5 | 30.1 |

current MLLMs struggle to reason about such spatial configurations across perspectives. The smallest discrepancy arises in Skill Evaluation, where even humans find domain-specific assessments (*e.g.*, judging basketball proficiency) challenging. This suggests that effective cross-view skill assessment requires MLLMs to integrate both domain knowledge and multi-view reasoning.

## 4.3 Can Reasoning Improve MLLM's Performance?

Prompting techniques [32, 67] have been shown to enhance the performance of MLLMs on various reasoning tasks [67, 83, 15, 55]. To examine whether similar reasoning prompts can improve the cross-perspective understanding, we sample 100 questions from each subtask and compare baseline zero-shot MCQ performance against CoT-augmented prompts. Following [32, 65], we append "Let's think step by step" to each prompt and evaluate four models representing diverse architectures and scales: LLaVA-OV-7B [35], Qwen2.5-VL-32B [76], Qwen2.5-VL-72B [62], and GPT-4o [29].

Figure 4 shows the accuracy difference ($Acc_{\text{CoT}} - Acc_{\text{baseline}}$) across all subtasks. CoT prompting degrades performance on most tasks. In particular, for the Person Matching and Action Matching subtasks, Qwen2.5-VL-72B [62], which already performs well without CoT, suffers a substantial drop in accuracy (20% and 19%). One reason for the performance drop is that the tasks in EgoExoBench demand that a model alternate between interpreting one video stream, translating that understanding to language, and then applying it to a second stream. Standard CoT cues, which focus purely on textual decomposition, disrupt this interleaved visual–linguistic workflow, leading to degraded performance.

Figure 5 provides a concrete example from Egocentric Wearer Identification with GPT-4o [29] under CoT prompting. The model correctly deduces from the first-person clip that the wearer stands next to someone in a blue jacket (highlighted in green). However, during the subsequent reasoning step, it misidentifies the person in the blue jacket as the camera wearer, rather than someone nearby (highlighted in red). This failure illustrates how purely text-oriented CoT reasoning can break the continuity of cross-view inference, underscoring the need for new prompting or architectural techniques that integrate visual and linguistic reasoning in tandem.

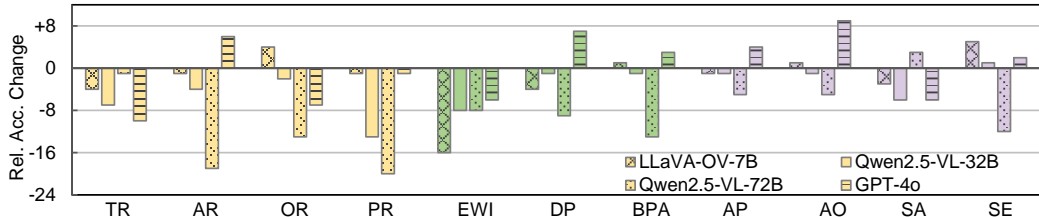

Figure 4: Relative performance changes after applying CoT. On average, CoT leads to a performance drop across tasks. This suggests that EgoExoBench cannot be effectively addressed by linguistic reasoning alone.

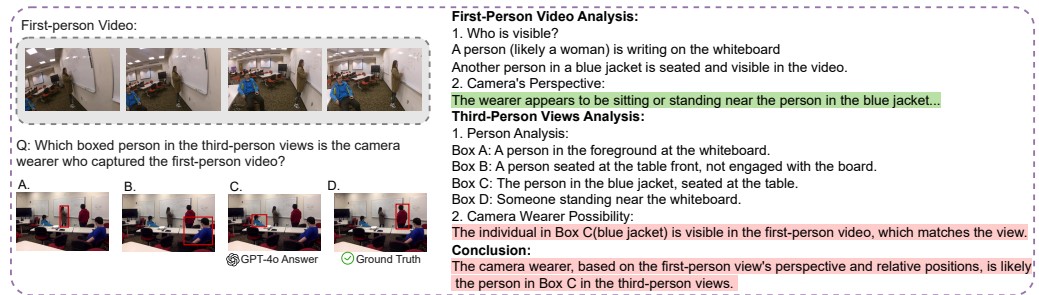

Figure 5: A failure case of GPT-4o with CoT prompting. The model fails to maintain consistency across cross-perspective visual-linguistic reasoning.

## 4.4 Can MLLMs Leverage Cross-Perspective Guidance?

To determine whether MLLMs can leverage additional cross-view context, we augment the Action Prediction and Skill Evaluation tasks with an additional video from a different perspective as a reference. For each question, the model receives both the primary clip (*e.g.*, egocentric for Action Prediction) and the reference clip (*e.g.*, exocentric), then selects the answer. Meanwhile, we perform an ablation by removing the reference clip and measuring the change in accuracy. The results are presented in Table 2. In the Action Prediction task, providing the reference video consistently boosts performance. For instance, Qwen2.5-VL [62] achieves a 8.2% gain. This improvement confirms the practical value of cross-perspective cues for better understanding actions. In contrast, results on the Skill Evaluation task are less consistent. While GPT-4o [29] and InternVL3 [85] show small performance differences, Qwen2.5-VL [62] experiences a 1.5% drop when the reference video is included. We hypothesize that this task's reliance on domain-specific expertise and subtle quality judgments, such as assessing proficiency in sports, outweighs the benefits of extra visual context. Under current model capabilities, cross-view input alone may be insufficient to enhance fine-grained skill assessment without deeper incorporation of domain knowledge.

Table 2: Ablation study on the use of reference videos.

|  | Action Prediction | | Skill Evaluation | |
|---|---|---|---|---|
|  | w.o./ref | w./ref | w.o./ref | w./ref |
| Qwen2.5-VL-72B | 31.7 | 39.9 (+8.2) | 31.4 | 29.9 (-1.5) |
| InternVL3-78B | 41.9 | 51.0 (+9.1) | 30.4 | 29.5 (-0.9) |
| GPT-4o | 26.1 | 27.7 (+1.1) | 26.5 | 26.3 (-0.2) |

## 4.5 What Makes the Egocentric Viewpoint Special?

To examine the uniqueness of egocentric viewpoints, we conduct exo–exo ablation studies across multiple tasks. In each task, the egocentric video in a QA pair is replaced with its synchronized third-person counterpart, enabling direct comparison between egocentric and exocentric settings. We use videos from LEMMA and EgoExo4D, which provide temporally aligned ego–exo recordings.

Table 3: Ego–Exo ablation study on ego–exo matching, with results grouped by scenario type.

| Model | View | Cooking | Household | Entertainment | Sport | Assembly |
|---|---|---|---|---|---|---|
| Qwen2.5-VL-7B | Ego–Exo | 47 | 41 | 46 | 27 | 25 |
| | Exo–Exo | 49 (+2) | 33 (-8) | 46 (0) | 27 (0) | 40 (+15) |
| Qwen2.5-VL-72B | Ego–Exo | 61 | 68 | 54 | 30 | 45 |
| | Exo–Exo | 58 (-3) | 56 (-12) | 46 (-8) | 36 (+6) | 60 (+15) |
| InternVL3-14B | Ego–Exo | 47 | 44 | 42 | 26 | 35 |
| | Exo–Exo | 44 (-3) | 41 (-3) | 50 (+8) | 34 (+8) | 43 (+8) |
| InternVL3-78B | Ego–Exo | 53 | 65 | 59 | 22 | 36 |
| | Exo–Exo | 52 (-1) | 51 (-14) | 48 (-11) | 30 (+8) | 48 (+12) |
| LLaVA-Video-7B | Ego–Exo | 37 | 48 | 30 | 22 | 20 |
| | Exo–Exo | 36 (-1) | 33 (-15) | 26 (-4) | 28 (+6) | 30 (+10) |
| LLaVA-OV-7B | Ego–Exo | 36 | 31 | 35 | 26 | 23 |
| | Exo–Exo | 35 (-1) | 27 (-4) | 26 (-9) | 29 (+3) | 33 (+10) |

Table 4: Ego–Exo ablation study on BPA, AO, SA, and SE tasks.

| Model | BPA | | AO | | SA | | SE | |
|---|---|---|---|---|---|---|---|---|
| | Ego–Exo | Exo–Exo | Ego–Exo | Exo–Exo | Ego–Exo | Exo–Exo | Ego–Exo | Exo–Exo |
| Qwen2.5-VL-7B | 29 | 37 (+8) | 26 | 28 (+2) | 37 | 39 (+2) | 27 | 32 (+5) |
| InternVL3-14B | 47 | 48 (+1) | 32 | 52 (+20) | 47 | 48 (+1) | 29 | 37 (+8) |
| LLaVA-Video-7B | 33 | 35 (+2) | 28 | 37 (+9) | 44 | 45 (+1) | 29 | 30 (+1) |
| Qwen2.5-VL-72B | 48 | 58 (+10) | 34 | 48 (+14) | 46 | 48 (+2) | 40 | 39 (-1) |
| InternVL3-78B | 38 | 51 (+13) | 32 | 46 (+14) | 52 | 54 (+2) | 30 | 45 (+15) |
| LLaVA-OV-7B | 25 | 25 (0) | 29 | 32 (+3) | 45 | 46 (+1) | 27 | 27 (0) |

Table 3 reports the results for the Ego–Exo Matching category. We group the results by scene type. Egocentric views exhibit advantages in Cooking, Household, and Entertainment scenarios, where fine-grained hand–object interactions and manipulation details are best observed from the actor's perspective. In contrast, exocentric views outperform in Sports and Assembly scenarios, which benefit from stable, wide-angle coverage of full-body motion and the overall workspace. We further extend this analysis to other tasks, as summarized in Table 4. Egocentric views are less effective for tasks that require global motion cues or temporal consistency(e.g., AO, SA, SE) due to issues like motion blur and limited field of view. These gaps underscore the need for improved egocentric representation learning and temporal modeling to enhance cross-view understanding in MLLMs.

## 5 Conclusion

EgoExoBench is the first large-scale benchmark explicitly designed to evaluate cross-view video understanding in multimodal LLMs. By aggregating paired egocentric–exocentric recordings and crafting over 7,300 high-quality multiple-choice questions across eleven subtasks, EgoExoBench probes three fundamental dimensions of ego–exo reasoning: semantic alignment, spatial correspondence, and sequence integration. Our extensive evaluation of both open- and closed-source models reveals that, despite strong single-view performance, current MLLMs struggle to bridge perspectives. Chain-of-thought prompting and additional cross-perspective guidance offer limited improvements, underscoring the need for novel architectures and training paradigms that can interleave visual and linguistic inference across multiple viewpoints. While the benchmark spans diverse tasks, it may not fully reflect the breadth of real-world ego–exo scenarios, which we leave for future work. We hope EgoExoBench will serve as a valuable resource to spur research on embodied agents and collaborative systems that require human-like cross-view intelligence.

## Acknowledgments and Disclosure of Funding

This work is supported by the National Natural Science Foundation of China (Grant No.62372223 and U24A20330), and in part by Nanjing University-China Mobile Communications Group Co., Ltd. Joint Institute under Grant NJ20250037, the National Key R&D Program of China (2022ZD0160201), JSPS KAKENHI JP25K24384, and Shanghai Artificial Intelligence Laboratory.

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

# Appendix

## A   Additional details and benchmark statistics

We first present additional statistics of EgoExoBench. Next, we detail the QA construction process for each task. Finally, we provide detailed information on the human evaluation performance.

### A.1   Benchmark Statistics

EgoExoBench is constructed from six publicly available ego–exo datasets and encompasses three task categories, comprising a total of 11 subtasks. Figure 6 summarizes the number of QA pairs contributed by each dataset for every subtask.

| Ego-Exo Matching | | | |
| --- | --- | --- | --- |
| **Task Matching(557)** | **Action Matching(828)** | **Object Matching(855)** | **Person Matching(497)** |
| LEMMA: 212
EgoExoLearn: 108
Ego-Exo4D: 237 | LEMMA: 612
EgoExoLearn: 216 | LEMMA: 855 | CVMHAT: 497 |

| Ego-Exo View Transition | | |
| --- | --- | --- |
| **Direction Prediction(300)** | **Egocentric Weaer Identification (534)** | **Body Part Action Understanding (786)** |
| Ego-Exo4D: 497 | TF2023: 534 | Ego-Exo4D: 325
EgoExoLearn: 461 |

| Ego-Exo Temporal Reasoning | | | |
| --- | --- | --- | --- |
| **Action Prediction(835)** | **Action Order(782)** | **Sequence Alignment (533)** | **Skill Evaluation(823)** |
| LEMMA: 420
EgoMe: 415 | LEMMA: 782 | Ego-Exo4D: 533 | EgoExoLearn: 444
Ego-Exo4D: 379 |

Figure 6: Overview of QA distribution in EgoExoBench.

### A.2   Ego-Exo Matching

Ego–Exo Matching evaluates a model's ability to associate semantically similar visual content across first- and third-person perspectives. Below, we detail the QA construction process for the task-, action-, person-, and object-level matching subtasks. Examples of QA instances for each subtask are illustrated in Figures 7 and 8.

**Task Matching.** We curate videos from the LEMMA [30], EgoExoLearn [27], and EgoExo4D [17] datasets. Since LEMMA [30] contains multi-agent scenarios involving multiple concurrent tasks, we include only videos that depict a single task to avoid ambiguity in task identification. For each QA pair, the ground-truth candidate video illustrates the same task as the query video but from a different viewpoint (e.g., egocentric vs. exocentric). To construct negative candidates, we avoid trivial distinctions based solely on environmental differences. In particular, distractor videos are selected from scenes of the same general type as the query video (e.g., all from kitchen environments). This design ensures that the model must rely on task-relevant visual cues rather than background differences. In total, we construct 212, 108, and 237 QA pairs from LEMMA [30], EgoExoLearn [27], and Ego-Exo4D [17], respectively.

**Action Matching.** We construct QA pairs using videos from the LEMMA [30] and EgoExoLearn [27] datasets. For LEMMA [30], we utilize the provided action annotations. The ground-truth (GT) video is drawn from the same source video as the query, capturing the same temporal segment from a different viewpoint. Negative candidates are selected from different temporal segments of the same video, corresponding to different actions. For EgoExoLearn [27], we use QA pairs from the

association benchmark. The GT video is directly provided. To construct negative candidates, we begin with the 20 candidate videos included in the benchmark. We first filter for candidates that share at least one verb or noun with the query. Then, we use SentenceTransformer [54] to compute the similarity between the query narration and each candidate's narration. The three most similar candidates (excluding the GT) are selected as negative options. This ensures distractors are relevant, maintaining task difficulty. In total, we construct 612 and 216 QA pairs from LEMMA [30] and EgoExoLearn [27], respectively.

**Object Matching.** We curate videos from the LEMMA [30] dataset. The interacted object associated with the person is extracted based on action annotations. Similar to the Action Matching subtask, for each query video (e.g., from the egocentric view), the ground-truth candidate is a temporally aligned clip from the corresponding third-person view. Negative candidates are sampled from the same third-person video but correspond to different time segments where the person interacts with different objects.

**Person Matching.** We construct data from the CVMHAT [19] dataset, which provides synchronized egocentric and top-view recordings across five outdoor scenes. To indicate the target person in each video, we overlay bounding boxes derived from the dataset's annotations. The query video is an egocentric clip captured by a specific individual, while all candidate videos are sourced from the corresponding top-view frame at the same timestamp. The difference among candidates lies in the identity of the person highlighted by the bounding box.

## A.3  Ego–Exo View Transition

The subtasks in Ego–Exo View Transition evaluate a model's ability to translate spatial information between first- and third-person perspectives. Figure 9 illustrates QA examples from each subtask. Below, we provide more details on the construction of QA pairs for each subtask.

**Egocentric Wearer Identification.** We adopt an annotation-derived strategy to construct QA pairs. We curate data from TF2023 [82] dataset, which provides synchronized egocentric–exocentric image pairs. Given an egocentric video, the goal is to identify the corresponding wearer in the third-person view. Candidates are constructed following the same method as in the Person Matching subtask. For each option, the target person is marked with a bounding box. To increase spatial complexity, we select only samples that contain four or more individuals in the scene.

**Direction Prediction.** We adopt a human-annotated strategy to construct the data. The process consists of five main steps: *(1) Video Selection.* We select synchronized multi-view videos from the Ego-Exo4D [17]. *(2) Action Segment Filtering.* We use the atomic descriptions annotations provided in Ego-Exo4D and retain only segments where the action is marked as visible in the egocentric view. We then apply Qwen2.5-32B [76] to determine whether the action description contains directional information (e.g., moving forward). Only segments with explicit directional content are kept. *(3) QA Annotation.* We hire three student annotators to create QA pairs based on the filtered segments. Each annotator accesses synchronized egocentric and third-person videos and is instructed to focus on the movement direction of the actor or the object being interacted with. If the action is unclear in the egocentric view or the movement direction cannot be judged from any third-person view, the sample is discarded. To construct question, the annotator selects a movement direction observed in one third-person view and writes a question referring to that direction. Movement descriptions are written in free-form natural language. Each question is designed to have exactly one correct answer among the candidates. *(4) Question Polishing.* We use Qwen2.5-32B [76] to refine the questions for clarity and consistency. *(5) Quality Assurance.* We hire an additional student, independent from the annotation process, to review each QA pair. The reviewer ensures that each question has only one correct answer, and the described direction accurately corresponds to the correct video.

**Body Part Action Understanding.** We adopt an LLM-based approach to construct QA pairs. Each question is derived from a pair of videos that share at least one verb or object. We input the textual descriptions of both videos into DeepSeek-V3 [40] and prompt it to generate a question that compares the actions, with an explicit focus on body parts involved. The prompting strategy is shown in Figure 10. To ensure the questions require visual information, we further use Qwen2.5-32B [76] to filter out those that can be answered without visual input. The filtering prompt is shown in Figure 11.

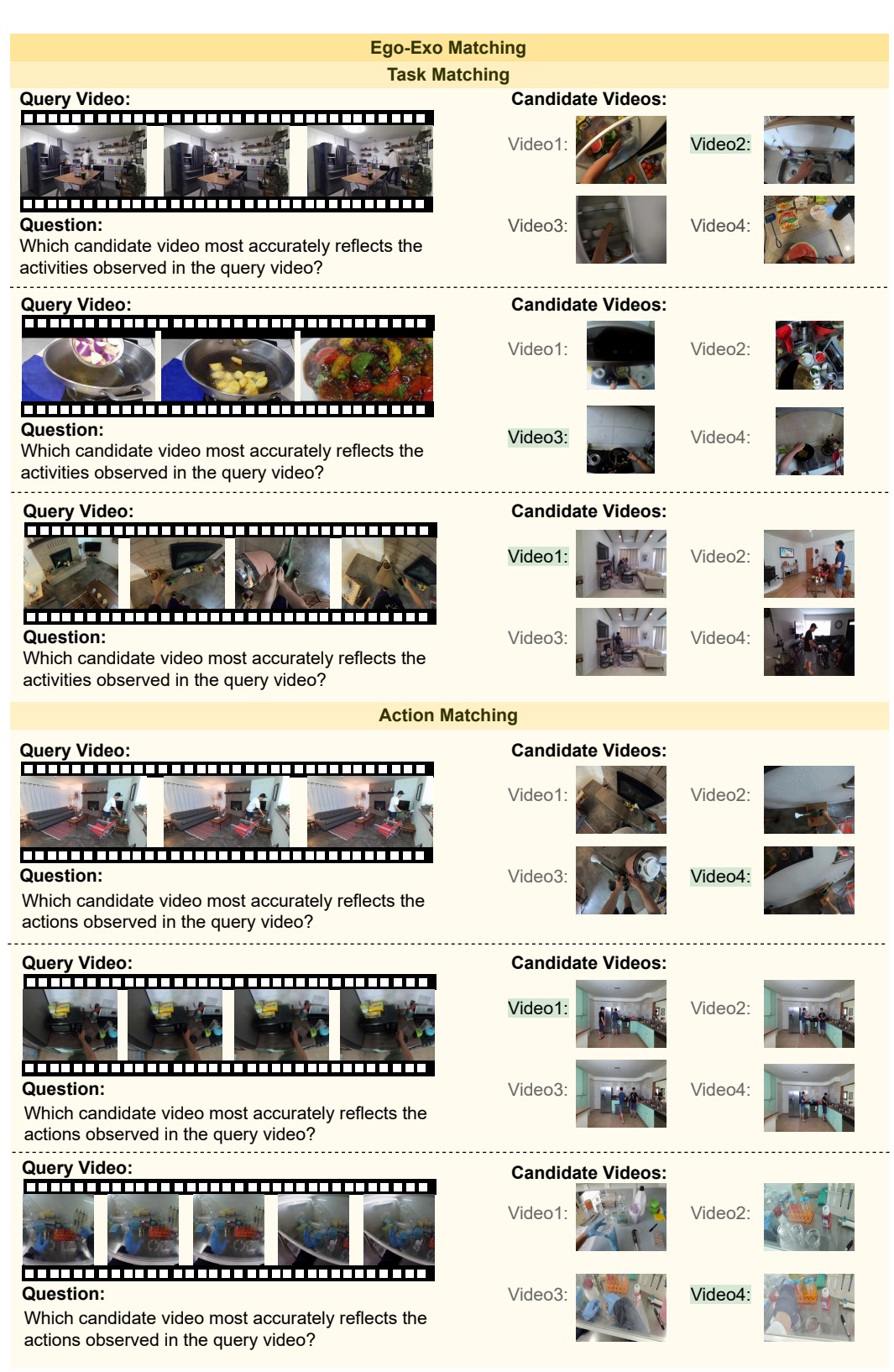

Figure 7: Examples on Ego–Exo Matching. Correct answers are highlighted in green.

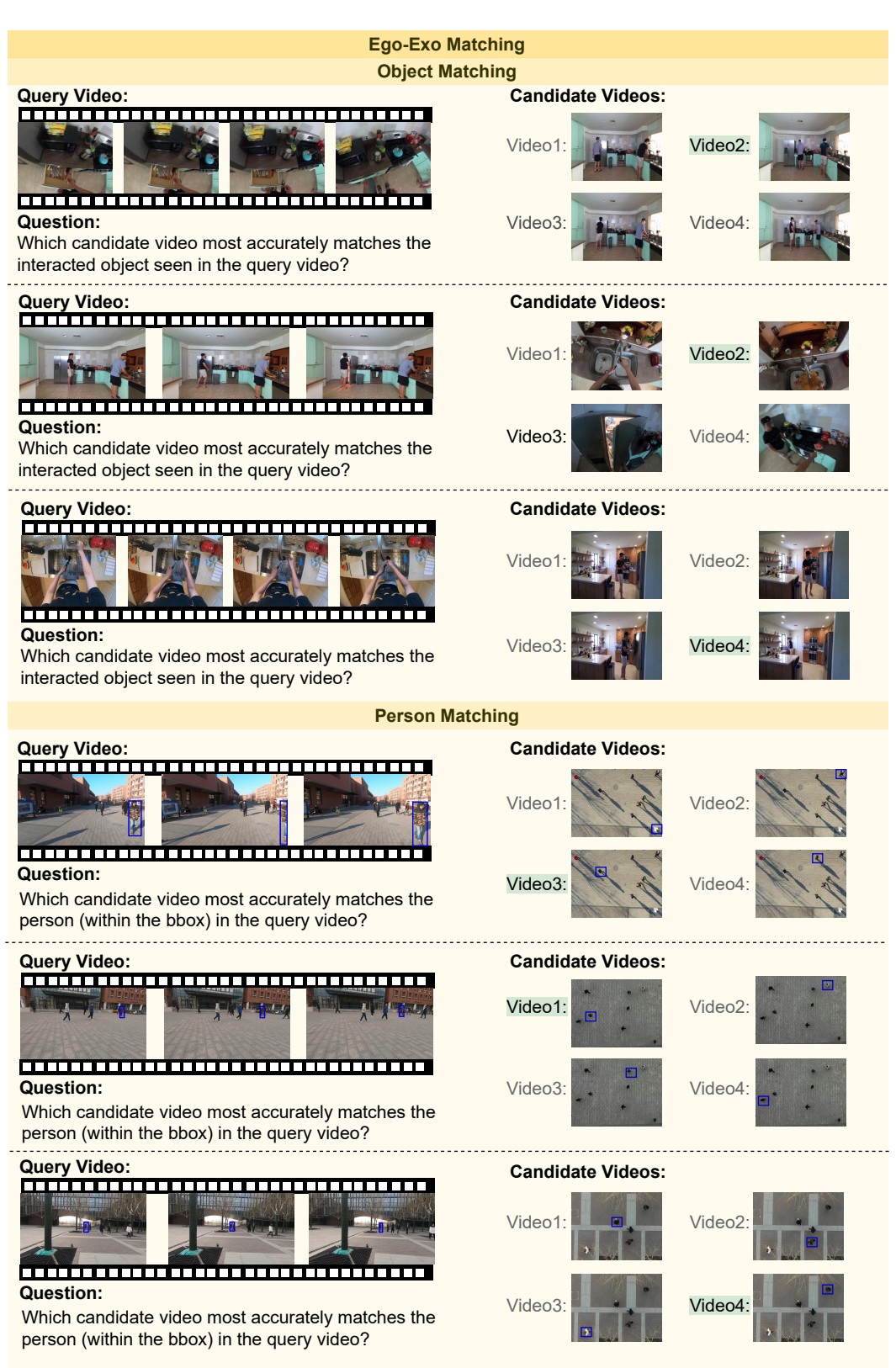

Figure 8: Examples on Ego–Exo Matching. Correct answers are highlighted in green.

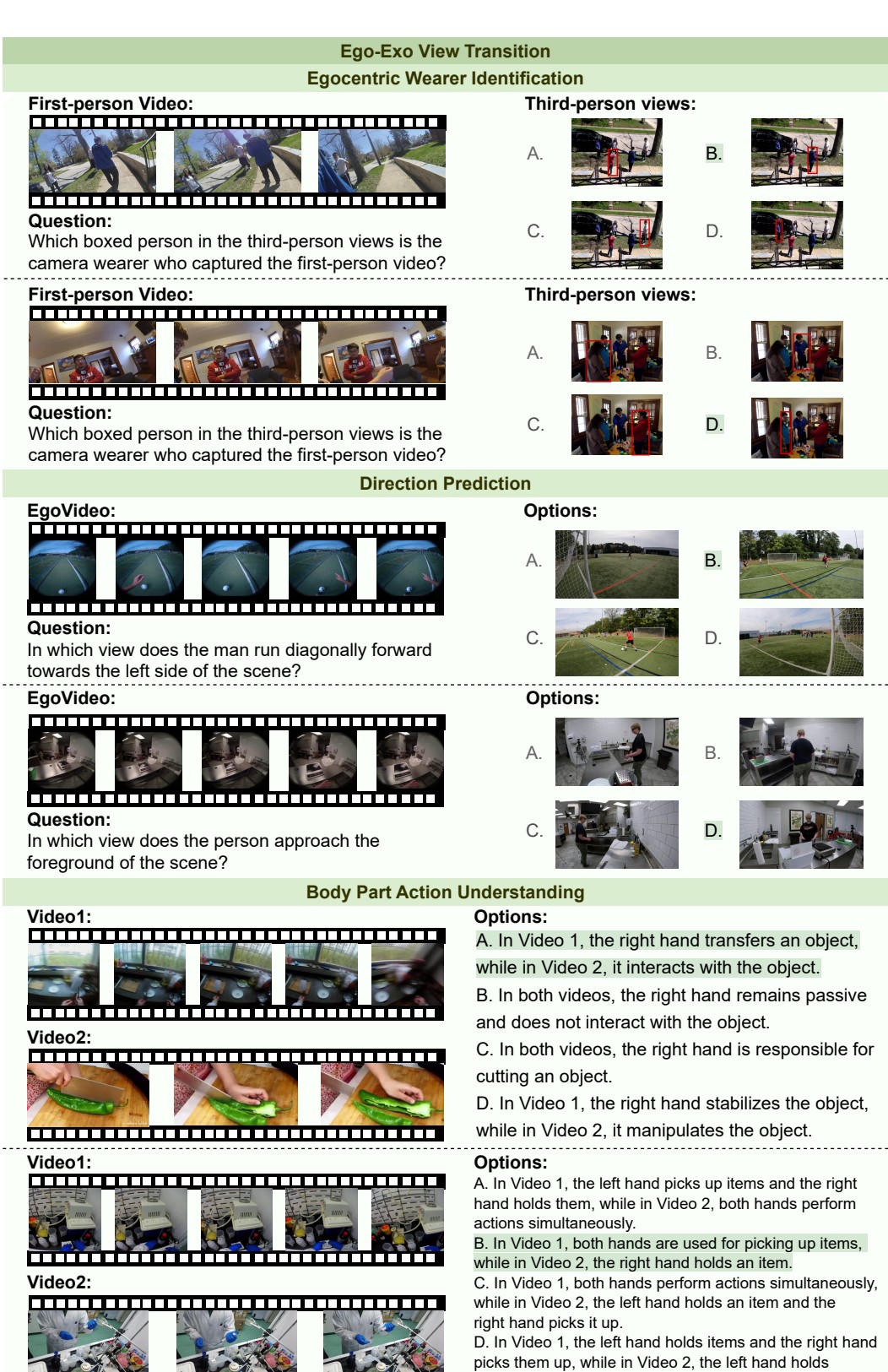

Figure 9: Examples on Ego–Exo View Transition. Correct answers are highlighted in green.

You will be provided with two text descriptions of different video scenes. Each description involves a person performing actions that engage specific body parts. Based on these descriptions, you are required to generate a set of question-and-answer (QA) pairs.

Requirements:
1. Each QA pair should include one question, four answer options, and one correct answer.
2. The question must clearly address details from both descriptions, particularly focusing on the roles of the body parts, without directly referencing specific actions or objects to prevent giving away hints.
3. Ensure there is one correct answer, and the other three options should be plausible but incorrect to increase the difficulty.
4. Use 'Video 1' to refer Video 1, use 'Video 2' to refer Video 2.
5. The output must be in JSON format as follows:
```json
{
"Question": "The question text goes here",
"Options": {
"A": "Option A text",
"B": "Option B text",
"C": "Option C text",
"D": "Option D text"
},
"Answer": "Correct option letter (e.g., 'A')"
}
```

Example Input:
Video 1: I hold the knife with my left hand and press the garlic with my right hand to slice it.
Video 2: Hold the chili pepper with your left hand and cut it into pieces with a knife in your right hand.

Output Example Based on the Above Input:
```json
{
"Question": "How do the tasks performed by the left and right hands differ in the two videos?",
"Options": {
"A": "In Video 1, the left hand holds the knife and the right hand slices; in Video 2, the left hand holds the chili and the right hand cuts.",
"B": "In Video 1, the left hand presses the garlic and the right hand holds the knife; in Video 2, the left hand cuts the chili and the right hand holds the knife.",
"C": "In Video 1, the left hand holds the knife and the right hand presses the garlic; in Video 2, the left hand holds the knife and the right hand cuts the chili.",
"D": "In Video 1, the left hand cuts the garlic and the right hand holds the knife; in Video 2, the left hand holds the chili and the right hand cuts."
},
"Answer": "A"
}
```

Video1: [description]
Video2: [description]

Figure 10: Prompt for QA generation in the Body Part Action Understanding subtask.

Carefully read the question and all the options, and based on the information provided, select and provide the most likely correct answer.
[Question]
[Options]
The answer is:

Figure 11: Prompt used to filter out QA pairs that can be correctly answered using text-only input.

## A.4 Ego–Exo Temporal Reasoning

Ego–Exo Temporal Reasoning evaluates a model's ability to align and infer event sequences across egocentric and exocentric video streams. Figures 16 and 17 present example QA pairs for each subtask. Below, we provide further details on the QA construction process for each subtask.

**Action Prediction.** We adopt an annotation-derived strategy to construct QA data from the LEMMA [30] and EgoMe [49] datasets. For LEMMA [30], we utilize the provided action annotations. To generate candidate answers, ten actions surrounding the current one in the egocentric sequence are selected, excluding the true next action. These are then input into Qwen2.5-32B [76], which selects the three most plausible distractors. The prompting strategy is illustrated in Figure 12. For EgoMe [49], we use fine-grained step annotations. The previous, current, and next actions in the egocentric sequence are given to DeepSeek-V3 [40] to generate three distractors. The corresponding prompt is shown in Figure 13. To ensure data quality, we retain only samples where the current action appears uniquely in the sequence, eliminating ambiguity in next-step prediction. Additionally, Qwen2.5-32B [76] is used to confirm that the distractors do not include the correct answer. Following the approach in the Body Part Action Understanding task, we also apply Qwen2.5-32B [76] to remove any QA pairs that can be answered correctly using text alone.

> For the current action, which of the following actions is most likely to be the next action?
> Current action: [action name]
> Actions: [action names], [action names], ...
> The answer is:

Figure 12: Prompt for generating negative options for Action Prediction questions constructed from the LEMMA dataset.

**Action Order.** We adopt an annotation-derived strategy to construct QA data based on the LEMMA [30] dataset. Each question instance consists of a pair of temporally adjacent action clips: one from the exocentric view and the other from the corresponding egocentric view. To ensure valid temporal relationships, we prompt Qwen2.5-32B [76] with the descriptions of the two actions to filter out pairs that do not exhibit a plausible temporal order. The prompting strategy is shown in Figure 14.

**Sequence Alignment.** We adopt an LLM-based strategy to construct QA data based on the Ego-Exo4D [17] dataset. For each video pair, we prompt DeepSeek-V3 [40] with the corresponding keystep annotations to generate QAs that highlight key differences in the action sequences. The prompting strategy is illustrated in Figure 15. For quality control, we use Qwen2.5-32B [76] to verify each option against the keystep annotations and discard QAs with multiple valid answers. Finally, as in the Action Prediction subtask, we filter out QAs that can be correctly answered using text input alone.

**Skill Evaluation.** We adopt an annotation-derived strategy to construct QA data using the EgoExoLearn [27] and Ego-Exo4D [17] datasets. For EgoExoLearn [27], we use exocentric demonstration videos as references. Then, we construct transitive ranking chains from pairwise skill annotations to derive candidate videos. For Ego-Exo4D [17], participant proficiency is categorized into Novice, Early Expert, Intermediate Expert, and Late Expert. We select Late Expert videos as references and sample candidates with varying proficiency levels from the same activity. In total, we curate 444 and 379 QA pairs from EgoExoLearn [27] and Ego-Exo4D [17], respectively.

## A.5 Mitigation of Background Bias.

We implement strategies to minimize background interference in our benchmark. All questions and candidate videos are drawn from the same scene types, preventing models from relying on superficial background cues. For distinguishing tasks, we introduce stricter, task-specific controls, summarized in Table 5. Overall, these measures ensure that each task evaluates the intended skill rather than extraneous visual features.

Task Description:
You are tasked with generating multiple-choice distractor options (negative options) for a next-action prediction task. The goal is to create plausible and reasonable options that could be mistaken for the correct next action.Each distractor must adhere to the following criteria:
Logical Consistency: All options must align with the sequence of actions described in the "previous" and "current" actions. They should follow naturally from the context provided.
Reasonableness: Each option must represent a potential next action that could reasonably occur in the given scenario. Avoid introducing actions that are unrelated, overly complex, or implausible.
Non-redundancy: Distractors must not duplicate the correct answer or each other. Each option should introduce a distinct possibility for the next action.
No Obvious Errors: Ensure that none of the distractors contain logical flaws, contradictions, or actions that clearly violate the context.

Input Format:
You will receive the following inputs:
Previous Actions: A description of the actions that occurred before the current moment.
Current Action: A description of the action currently being performed.
Correct Next Action: The correct next action in the sequence (this will not be included in your output).

Output Format:
Generate 3 plausible distractor options that meet the criteria above. Each option should be concise and clearly describe a potential next action.

Example Input:
Previous Actions: "On the white table in front, there are two silver mice and a screwdriver. The left hand picks up the mouse on the left, and the right hand picks up the screwdriver on the right."
Current Action: "The right hand turns the screwdriver to loosen the screws on the mouse."
Correct Next Action: "The right hand rotates the screwdriver to tighten the screws on the mouse."

Example Output:
"The left hand steadies the mouse while the right hand inspects the loosened screws."
"The person pauses to examine the screwdriver for any signs of damage before continuing."
 "The right hand uses the screwdriver to start adjusting another screw on the mouse."

Previous Actions: [action name], [action name], ...
Current Action: [action name]
Correct Next Action: [action name]

Figure 13: Prompt for generating negative options for Action Prediction questions constructed from the EgoMe dataset.

You are an assistant that understands the logical sequence of everyday actions. Please determine whether there is a clear cause-effect or sequential dependency between the following two actions:
Action1: [action name]
Action2: [action name]
Rules:
Respond with "True" if one action must precede the other.
Respond with "False" if the actions can be performed in any order or independently.
The answer is:

Figure 14: Prompt for filtering action pairs in the Action Order subtask to ensure plausible temporal relationships.

You are a professional test designer tasked with creating a multiple-choice question based on the following two video action sequences.

Design Requirements:
Question: Clearly focus on the differences in the order of actions or the presence/absence of specific actions between Video 1 and Video 2.

Options: Provide four logically clear options (A, B, C, D) directly sourced from the given action sequences. Each option must closely address the action sequences or specific actions within the two videos.
Correct Option: This option should precisely identify one specific difference between the two videos, whether it's about the sequence of actions or the presence/absence of an action.
Incorrect Options: These options must contradict the video content by:
1. Providing an incorrect sequence of actions;
2. Making false statements about the presence or absence of specific actions;
Accuracy: Avoid vague or overly subjective questions. Do not introduce actions or details that are not present in the provided sequences.
Naming Convention: Refer to the first video as "Video 1" and the second video as "Video 2".

The output must be in JSON format as follows:
```json
{
  "Question": "The question text goes here",
  "Options": {
    "A": "Option A text",
    "B": "Option B text",
    "C": "Option C text",
    "D": "Option D text"
  },
  "Answer": "Correct option letter (e.g., 'A')"
}
```

Video 1 Action Sequence: []
Video 2 Action Sequence: []

Figure 15: Prompt for QA generation in the Sequence Alignment subtask.

## A.6 Human Performance Evaluation.

To estimate human-level performance on EgoExoBench, we randomly sample 30 QAs from each subtask, resulting in a total of 330 questions. Two graduate students are invited to complete this subset. To avoid annotation bias, the evaluators do not participate in the data construction process. Each evaluator receives the same input as MLLMs, including the question, options, and corresponding videos. Evaluators are instructed to answer all questions to the best of their ability. They are allowed to pause, replay, and watch the videos multiple times without time constraints. We report the average accuracy of the evaluators as the human performance baseline for this evaluation subset.

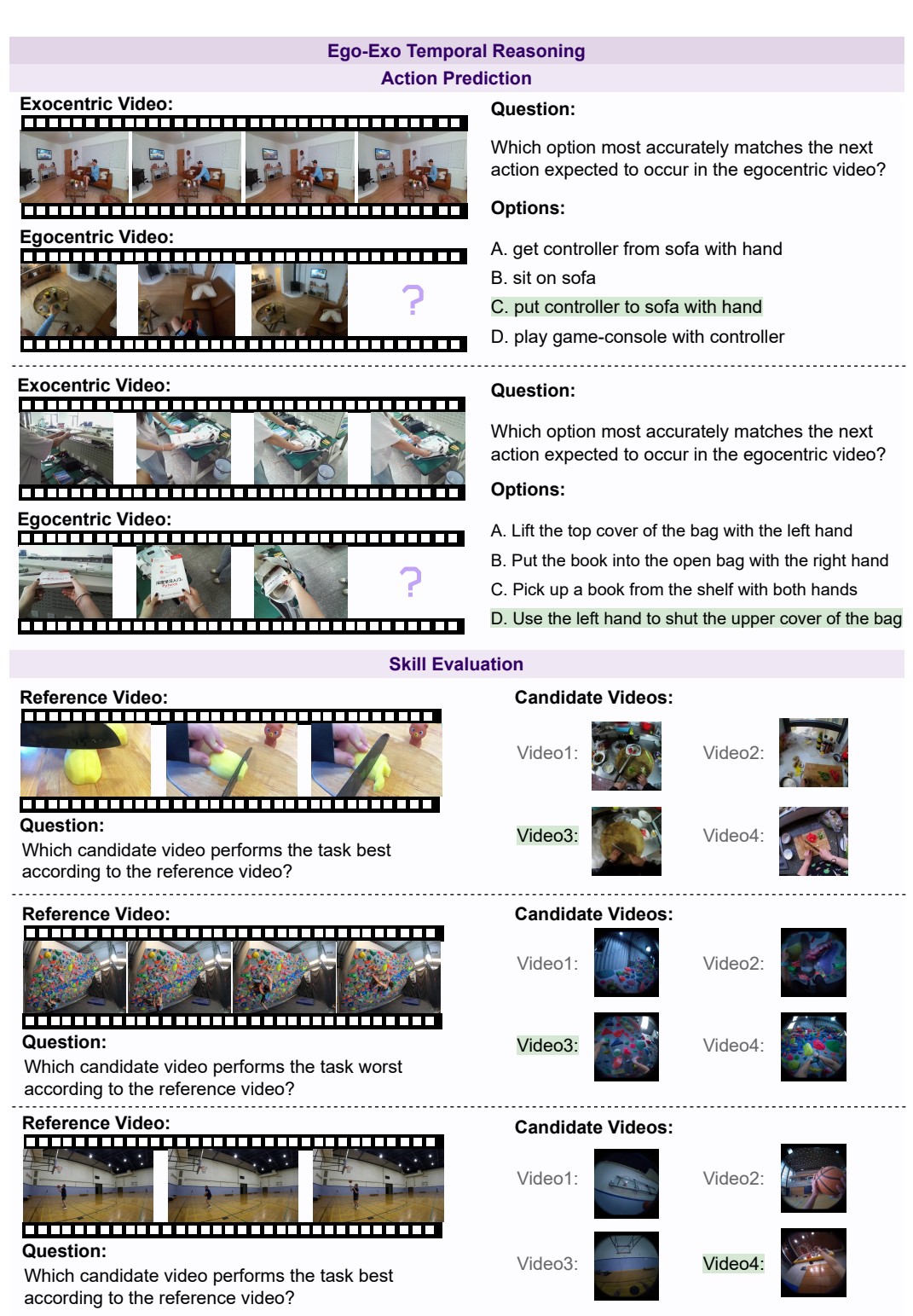

Figure 16: Examples on Ego–Exo Temporal Reasoning. Correct answers are highlighted in green.

**Ego-Exo Temporal Reasoning**

**Action Order**

**Video1:**

**Video2:**

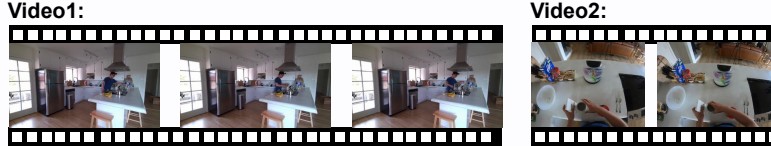

**Question:**

What is the correct temporal relationship between the actions in these two videos?

A. Actions in Video 1 occur before actions in Video 2
B. Actions in Video 2 are part of the sequence of actions in Video 1
C. Actions in Video 2 occur before actions in Video 1
D. Actions in Video 1 are part of the sequence of actions in Video 2

**Video1:**

**Video2:**

**Question:**

What is the correct temporal relationship between the actions in these two videos?

A. Actions in Video 2 occur before actions in Video 1
B. Actions in Video 1 are part of the sequence of actions in Video 2
C. Actions in Video 2 are part of the sequence of actions in Video 1
D. Actions in Video 1 occur before actions in Video 2

**Sequence Alignment**

**Video1:**

**Video2:**

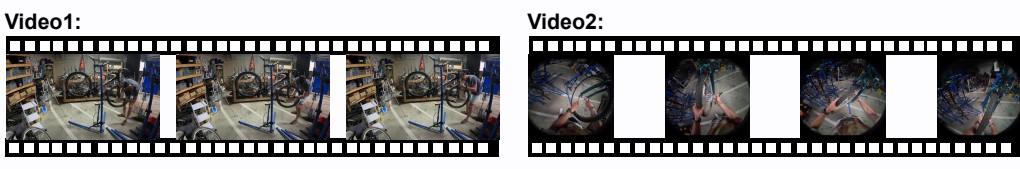

**Question:**

Which of the following statements accurately describes the difference in action sequences between Video 1 and Video 2?

A. Both videos show the same sequence of actions without any differences
B. Video 1 starts with rolling the wheel, while Video 2 starts with tightening both axle nuts
C. Video 1 includes tightening the brake cable to the rear axle, while Video 2 does not
D. Video 2 includes pushing the level inward and turning the axle clockwise, while Video 1 does not

**Video1:**

**Video2:**

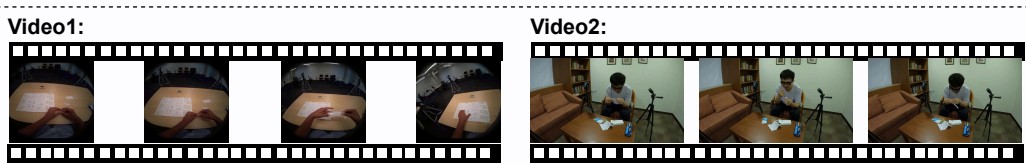

**Question:**

Which of the following statements accurately describes the difference in action sequences between Video 1 and Video 2?

A. Video 1 includes reading the instructions, while Video 2 does not
B. Video 2 includes arranging test material, while Video 1 does not
C. Video 1 includes rotating and swirling the swab, while Video 2 does not
D. Video 2 includes carefully opening the test tube seal paper, while Video 1 does not

Figure 17: Examples on Ego–Exo Temporal Reasoning. Correct answers are highlighted in green.

Table 5: Task-specific strategies to mitigate background interference. *Scene type* indicates high-level semantic categories (e.g., "kitchen", "lab"), *Environment* denotes specific locations (e.g., kitchen 1 vs. kitchen 2), *Same environment* refers to videos from the same physical site but different viewpoints, and *Same video* means options from the same original file.

| SubTask Type | Source | Scene Consistency | Remarks |
|---|---|---|---|
| OR, AR(LEMMA) | Same video | No background variation | All candidates from same video; GT-candidate from temporally-aligned clip; negatives differ temporally |
| PR, EWI | Same video | No background variation | All candidates from same video and clip; differ only by bounding boxes |
| AR (EgoExoLearn) | Same scene type, diff. env. | Candidates from same scene type (e.g., kitchens, labs) but diff. locations | Tests generalization across environments |
| DP, AO | Same env., diff. viewpoints | No background variation | All videos from same physical environment; vary only by viewpoint |
| TR | Same scene type, diff. env. | All videos from same scene type | Models rely on high-level task semantics, not background |
| SA, AP, SE | Same scene type | All videos from same scene type | Evaluates ability to align or predict actions/skills across similar environments |

# B Experiments

We begin by designing an experiment to evaluate whether current MLLMs can accurately identify the boundaries between multiple concatenated video inputs. We then present additional results on the EgoExoBench benchmark and compare model performance on EgoExoBench with that on other video understanding benchmarks. Finally, we provide experimental details and qualitative examples related to the Chain-of-Thought (CoT) prompting strategies and the ablation study on the impact of using reference videos.

## B.1 Preliminary Study: Can MLLMs Distinguish Multiple Video Inputs?

Before conducting the main experiments, we verify whether MLLMs can differentiate multiple video inputs as distinct streams. This capability is critical, as EgoExoBench tasks require to compare or relate information across several videos simultaneously. To this end, we design an identical video pair identification task. Each question presents five video clips, two of which are exact duplicates. The model's objective is to identify the matching pair. An example of this task is illustrated in Figure 18.

We curate video clips from Ego-Exo4D [17], including both egocentric and exocentric views. We construct a total of 100 question instances for evaluation. For each video, we uniformly sample 8 frames as input. Accuracy is used as the evaluation metric, where a response is considered correct only if the model precisely identifies the matching video pair. The expected accuracy under random guessing is 10%.

Table 6: Model performance on the preliminary experiment.

| Model | Acc. |
| --- | --- |
| Qwen2.5-VL-7B | 97 |
| InternVL3-8B | 85 |
| LLaVA-Video-7B | 77 |
| NVILA-8B | 78 |
| EgoGPT | 79 |
| Claude-3.7-Sonnet | 89 |

As shown in Table 6, all evaluated models achieve an accuracy above 77%, substantially exceeding random guess performance. This result confirms that current MLLMs are generally capable of distinguishing between multiple video inputs. This finding supports the feasibility of the results in our main experiments.

## B.2 Main Results

We additionally evaluate Gemini 2.5 Pro [12], Qwen2.5-Omni [74], Ola [44], on EgoExoBench. As shown in Table 7, Gemini 2.5 Pro [12] achieves the highest overall performance, with an average accuracy of 51.7%, outperforming the second-best model, GPT-4o-mini [47], by 3.7%. We also report the performance of MLLMs on several widely used video understanding benchmarks. As shown in Table 8, open-source models such as Qwen2.5VL-72B [62] and InternVL3-78B [85] outperform the closed-source GPT-4o [29] on certain benchmarks. Building on these overall results, we further highlight the importance of high-resolution visual encoding. Preserving fine-grained visual details is critical for tasks requiring precise spatial understanding, such as Ego–Exo View Transition and parts of Ego–Exo Matching. Models like InternVL3 and Qwen2.5-VL, which leverage dynamic resolution strategies to handle high-resolution inputs, consistently outperform those relying on fixed low-resolution inputs. This capability is especially beneficial for tasks like Egocentric Wearer Identification (EWI) and Body Part Action Understanding (BPA). Temporal modeling also plays a key role. Performance in the Ego–Exo Temporal Reasoning category correlates with the sophistication of a model's temporal encoding. Qwen2.5-VL's architecture introduces an absolute temporal encoding strategy. This provides the model with a physical sense of temporal flow and duration, a distinct advantage over simpler sequential encoding that merely counts frames. This architectural choice likely explains Qwen2.5-VL's superior performance on Action Ordering (AO).

**[Task Instruction]**
You will be provided with 5 videos. Please carefully analyze the content of each video and identify the two videos that are most likely to be identical.

**Video1:**

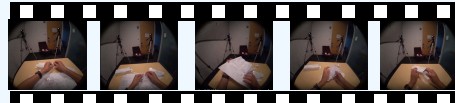

**Video2:**

**Video3:**

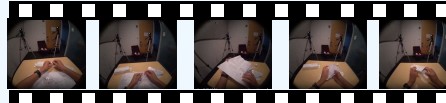

**Video4:**

**Video5:**

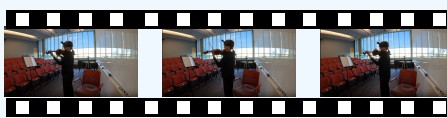

**[Responce Format]**
Output your answer in the format VideoX-VideoY, indicating the two videos you believe are most similar. The answer is:

**GT: Video1-Video3**

Figure 18: Example of the identical video pair identification task proposed in the preliminary experiment.

Table 7: Performance of various MLLMs on EgoExoBench. The best result among all models is highlighted in bold, and the second-best is underlined.

| | Avg. | Matching | | | | View Transition | | | Temporal Reasoning | | | |
|---|---|---|---|---|---|---|---|---|---|---|---|---|
| | | TR | AR | OR | PR | EWI | DP | BPA | AP | AO | SA | SE |
| Qwen2.5-VL-72B | 44.7 | 51.0 | 43.5 | 56.6 | 49.7 | 56.7 | 37.0 | **48.1** | 39.9 | 33.6 | 46.1 | 29.9 |
| InternVL3-78B | 41.4 | 50.6 | 37.3 | 48.9 | 39.0 | 46.9 | 32.3 | 38.0 | **51.0** | 31.5 | 50.6 | 29.5 |
| Claude-3.7-Sonnet | 31.3 | 33.4 | 33.0 | 35.1 | 30.6 | 37.8 | 30.0 | 33.3 | 26.0 | 28.5 | 30.2 | 26.4 |
| GPT-4o | 38.5 | 52.9 | 44.6 | 57.4 | 48.0 | 41.6 | 24.1 | 37.1 | 27.7 | 26.1 | 37.4 | 26.3 |
| GPT-o4-mini | 48.0 | **65.3** | 45.2 | 56.8 | 53.7 | 73.8 | 30.0 | 40.0 | 46.4 | **35.9** | 51.5 | 30.1 |
| Gemini 2.5 Pro | **51.7** | 63.1 | **52.1** | **67.3** | **56.8** | **76.0** | **38.0** | 47.7 | 50.8 | 33.9 | 49.3 | **33.4** |

Table 8: Comparison of MLLMs performance across EgoExoBench and existing video understanding benchmarks.

| | Video-MME | MLVU | LongVideoBench | CG-Bench | Egoschema | EgoExoBench |
|---|---|---|---|---|---|---|
| Qwen2.5-Omni-7B | 64.3/72.4 | - | - | - | 68.6 | 30.0 |
| Ola | 68.4/- | - | 61.4 | - | - | 28.0 |
| Qwen2.5-VL-7B | 65.1/71.6 | 70.2 | 56.0 | - | 65.0 | 32.8 |
| LLaVA-OV-7B | 58.2/- | 64.7 | 56.4 | 31.1/43.2 | 60.1 | 29.5 |
| LLaVA-Video-7B | 46.5/- | - | 43.5 | - | 57.3 | 31.2 |
| InternVL3-8B | 66.3/68.9 | 71.4 | 58.8 | 38.6/55.2 | - | 31.3 |
| InternVL3-14B | 70.4/73.0 | 73.3 | 63.9 | 44.1/60.6 | - | 35.1 |
| Qwen2.5-VL-72B | **73.3/79.1** | 74.6 | 60.7 | - | **76.2** | **44.7** |
| InternVL3-78B | 72.7/75.7 | **79.5** | 65.7 | **48.4/65.3** | - | 41.4 |
| GPT-4o | 71.9/77.2 | 64.6 | **66.7** | 41.8/58.3 | 72.2 | 38.5 |

Given a prediction for a multiple-choice question, directly extract the selected answer while skipping the reasoning process. If the prediction explicitly chooses option (e.g., A, B, C, D), return the corresponding letter. If the prediction does not specify a choice or indicates that none of the options are correct, return None.
[Prediction]

Figure 19: Prompt for extracting the final answer from CoT prediction.

## B.3 Experimental Details of CoT Prompting

As described in the main paper, we evaluate the effectiveness of Chain-of-Thought (CoT) prompting on our benchmark. Following [77], we append the phrase "Let's think step by step." to each question. The decoding parameters are configured with temperature set to 0, top-p to 1, and top-k to 1. To extract the final answer from the model's output, we employ Qwen2.5-32B [76] as a parser to explicitly extract the answer. The prompt design is illustrated in Figure 19.

**Case Analysis.** We present Chain-of-Thought (CoT) reasoning examples to illustrate how the model interprets videos and infers answer. Figure 20 shows a successful case of GPT-4o [29] on the Task Matching subtask. The model reasons by integrating environmental context and key actions of individuals. It identifies critical behaviors across videos, infers the underlying tasks, and compares candidate videos with the query video to select answer. Figure 21 illustrates a successful example on the Sequence Alignment subtask. The model first provides a step-by-step description of the action sequence for each video. Notably, it attends to the order in which participants interact with key objects (e.g., test tubes, testing plates). Furthermore, it distinguishes between fine-grained actions such as covering and unwrapping. This detailed understanding of action semantics and temporal ordering enables the model to effectively compare sequences across videos and infer the correct answer. Figure 22 presents a failure case of GPT-4o on the Body Part Action Understanding subtask. While GPT-4o successfully identifies the actions performed in both videos, it incorrectly attributes the action to the right hand in the egocentric video. In contrast, it correctly identifies the corresponding body part in the exocentric video. This suggests that GPT-4o still struggles to accurately localize left–right body part actions across different viewpoints. Figure 23 presents a failure case of GPT-4o on the Person Matching subtask. The model demonstrates inconsistency in the reasoning process. During the analysis of the query video, it infers that the target person is positioned at a distance from others. However, it subsequently selects an individual who is positioned closer to other people, contradicting its earlier observation. This inconsistency suggests that the model struggles to maintain reasoning continuity across cross-view inputs.

## B.4 Experimental Details on Reference Video Usage.

In the main paper, we evaluate whether models can effectively leverage cross-view information in the Action Prediction and Skill Evaluation subtasks. To assess the contribution of the reference video, we conduct an ablation study by removing it while retaining the original input video. To ensure fair comparison, we keep the input prompts as consistent as possible, modifying only the necessary components to reflect the absence of the reference video. All other experimental settings remain unchanged. Figure 24 and Figure 25 show the task prompts for the Action Prediction and Skill Evaluation subtasks, respectively. Figure 26 presents examples from the Action Prediction subtask, comparing GPT-4o's predictions with and without the reference video. Figure 27 shows similar comparisons for the Skill Evaluation subtask.

**Task Matching**

**Query Video:**

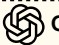

**Candidate Videos:**

Video1: 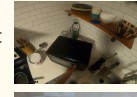    Video2: 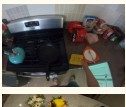

Video3:    Video4:

**Question:**
Which candidate video most accurately reflects the activities observed in the query video? Let's think step by step.

---

**GPT-4o Prediction:**

To determine which video closely matches the query video, let's analyze the query video step by step:

**Query Video Analysis:**

Setting: The video takes place in a kitchen.

Actions:

One person initially walks across the kitchen.

The second person is seen standing, then goes to the refrigerator.

The second person opens the refrigerator to take something out.

This person then proceeds to the counter and engages in food preparation near or at the stove.

**Comparing Candidate Videos:**

Let's examine each candidate video:

**Video 1:**

Setting: Different kitchen setup with a dark theme.

Actions: Involves actions like taking items from the refrigerator and using the microwave, which is not aligned with the query.

**Video 2:**

Setting: Similar kitchen setup to the query.

Actions:

A person goes to the refrigerator.

The person then uses the stove with utensils similar to those in the query, aligned with the sequence of opening the fridge, taking items, and preparing something on the stove.

**Video 3:**

Setting: Kitchen setup similar to the query.

Actions: Focuses on cutting a watermelon, which is not present in the query video.

**Video 4:**

Setting: Wooden kitchen cabinets, different style from query.

Actions: Involves using the refrigerator and pouring a drink. Does not involve using stove or food preparation activities.

**Conclusion:**

Video 2 fits the actions and setting closest to the query video, with the refrigerator and stove being used in a sequence consistent with the query. Thus, Video 2 most accurately reflects the activities observed in the query video. ✓

Figure 20: A successful case of GPT-4o's CoT reasoning on the Task Matching subtask. The model compares key actions across videos to identify the matching candidate. Key actions are highlighted in green.

**Sequence Alignment**

**Video1:**

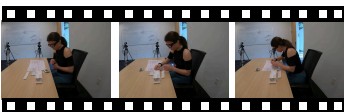

**Video2:**

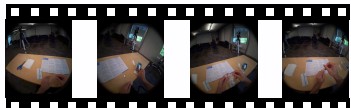

**Question:**
What is the key difference in the action sequences between Video 1 and Video 2? Let's think step by step.
**Options:**
A. In Video 1, the test tube is covered after unwrapping the testing plate, whereas in Video 2, the test tube is covered before unwrapping the test tube cap.
B. In Video 1, the test tube is covered before unwrapping the testing plate, whereas in Video 2, the test tube is covered after unwrapping the test tube cap.
C. In Video 1, the swab is swirled after covering the test tube, whereas in Video 2, the swab is swirled before removing it from the testing tube.
D. In Video 1, the testing plate is unwrapped before swirling the swab, whereas in Video 2, the testing plate is unwrapped after covering the test tube.

- - - - - - - - - - - - - - - - - - - - - - - - - - - - - - - - - - - - - - - - - - - - - - - - - - -

**GPT-4o Prediction:**

To determine the key difference in the action sequences, let's analyze each video step-by-step based on the provided videos:

Video 1:
1. The person appears to be preparing materials on the table.
2. They seem to be handling a test tube, possibly doing a covering or sealing action.
3. There's a focus on opening or handling some instruction or paper.
4. The testing plate is being unwrapped and possibly manipulated.

Video 2:
1. The person is seen handling materials on a table.
2. They unwrap the testing plate and handle the test tube.
3. After unwrapping, they perform actions with the test tube, such as covering it.

Conclusion:

After reviewing the sequences, Option B correctly describes the actions. 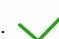

Figure 21: A successful case of GPT-4o's CoT reasoning on the Sequence Alignment subtask. The model captures the order of interactions with key objects and differentiates fine-grained actions, enabling accurate cross-video sequence comparison. Key reasoning steps are highlighted in green.

## Body Part Action Understanding

**Video1:**

**Video2:**

**Question:**
How do the roles of the left and right hands differ between Video 1 and Video 2? Let's think step by step.

**Options:**

A. In Video 1, the left hand handles seasoning; in Video 2, the left hand supports the pot and the right hand handles scooping.

B. In Video 1, the right hand handles seasoning; in Video 2, the right hand handles scooping and the left hand supports the pot.

C. In Video 1, the right hand handles seasoning and the right hand assists; in Video 2, the right hand supports the pot and the left hand handles scooping,

D. In Video 1, the left hand handles seasoning; in Video 2, the left hand handles scooping and the right hand supports the pot.

---

**GPT-4o Prediction:**

To determine the roles of the hands in each video, we analyze their actions separately.

**Video 1:**

The images show someone cooking over a stove. The right hand is adding seasoning from a packet to the pan. There is no clear indication that the left hand is supporting the pot, but it appears to assist during the process.

**Video 2:**

The cook is preparing a dish and serving it onto a plate. The left hand is holding and supporting the pan. The right hand is using a utensil to scoop food onto the plate.

**Comparison of hand roles:**

In Video 1, the right hand handles seasoning, and the left hand assists.

In Video 2, the right hand handles scooping, and the left hand supports the pot.

Based on this analysis, the correct option is: B. ✗

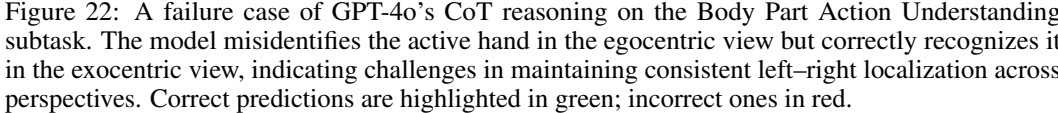

Figure 22: A failure case of GPT-4o's CoT reasoning on the Body Part Action Understanding subtask. The model misidentifies the active hand in the egocentric view but correctly recognizes it in the exocentric view, indicating challenges in maintaining consistent left–right localization across perspectives. Correct predictions are highlighted in green; incorrect ones in red.

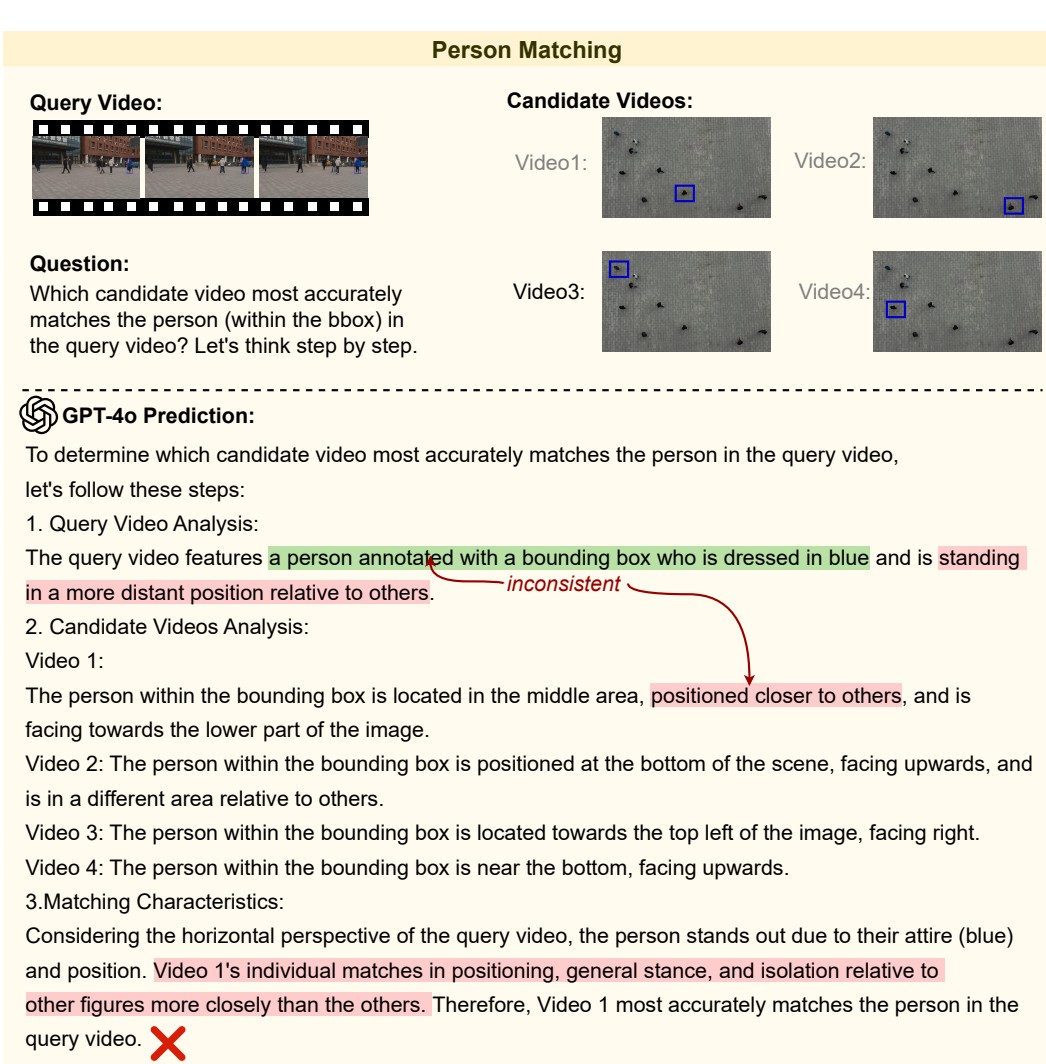

Figure 23: A failure case of GPT-4o on the Person Matching subtask. The model infers the target person is distant from others in the query video but selects a candidate positioned closer to others, revealing inconsistency in cross-view reasoning. Correct reasoning steps are highlighted in green. Inconsistent reasoning steps are highlighted in red.

**Prompt for Action Prediction with Reference Video**

You are provided with a third-person perspective (Exocentric) video and a first-person perspective (Egocentric) video. Leverage the exocentric video as the gold standard for understanding the correct order and details of action sequences. Your goal is to predict the next action that will occur in the egocentric video based on the sequence of actions observed in the exocentric video.

[Exocentric Video]

[Egocentric Video]

Question:

Which option most accurately matches the next action expected to occur in the egocentric video, according to the sequence established by the exocentric video?

Options:

A. [Action Name]    B. [Action Name]    C. [Action Name]    D. [Action Name]

---

**Prompt for Action Prediction without Reference Video**

You are provided with a first-person perspective (Egocentric) video. Your goal is to predict the next action that will occur in the egocentric video based on the sequence of actions observed in the egocentric video.

[Egocentric Video]

Question:

Which option most accurately matches the next action expected to occur in the egocentric video?

Options:

A. [Action Name]    B. [Action Name]    C. [Action Name]    D. [Action Name]

Figure 24: Prompts for the Action Prediction subtask, with and without the reference video.

---

**Prompt Example for Skill Evaluation with Reference Video**

You are a professional action skills evaluation assistant. Please watch a reference video performed by an expert, which demonstrates the correct technique and execution of the activity. Next, evaluate four different candidate videos (labeled A, B, C, and D). Each video features a different individual performing the same type of activity. Your task is to compare each candidate's performance against the expert's demonstration to evaluate their proficiency level.

[Reference Video]

Question:

Which candidate video performs the task best according to the reference video?

Candidates:
A. [Video]    B. [Video]    C. [Video]    D. [Video]

---

**Prompt Example for Skill Evaluation without Reference Video**

You are a professional action skills evaluation assistant. Please evaluate four different candidate videos (labeled A, B, C, and D). Each video features a different individual performing the same type of activity. Your task is to compare each candidate's performance to evaluate their proficiency level.

Question:

Which candidate video performs the task best according to the reference video?

Candidates:

A. [Video]    B. [Video]    C. [Video]    D. [Video]

Figure 25: Prompts for the Skill Evaluation subtask, with and without the reference video.

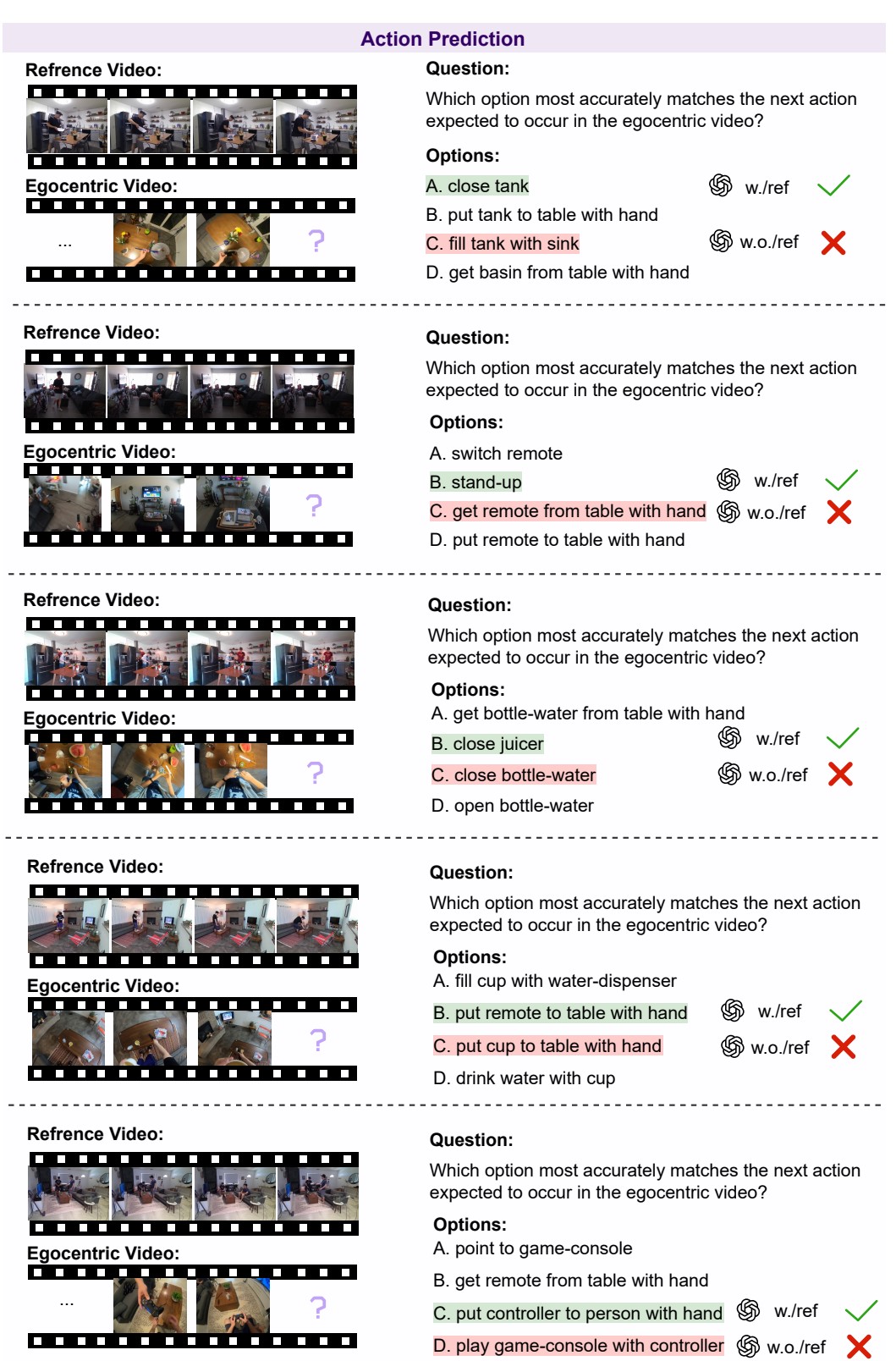

Figure 26: GPT-4o predictions on the Action Prediction subtask with and without the reference video. Correct predictions are highlighted in green; incorrect ones in red.

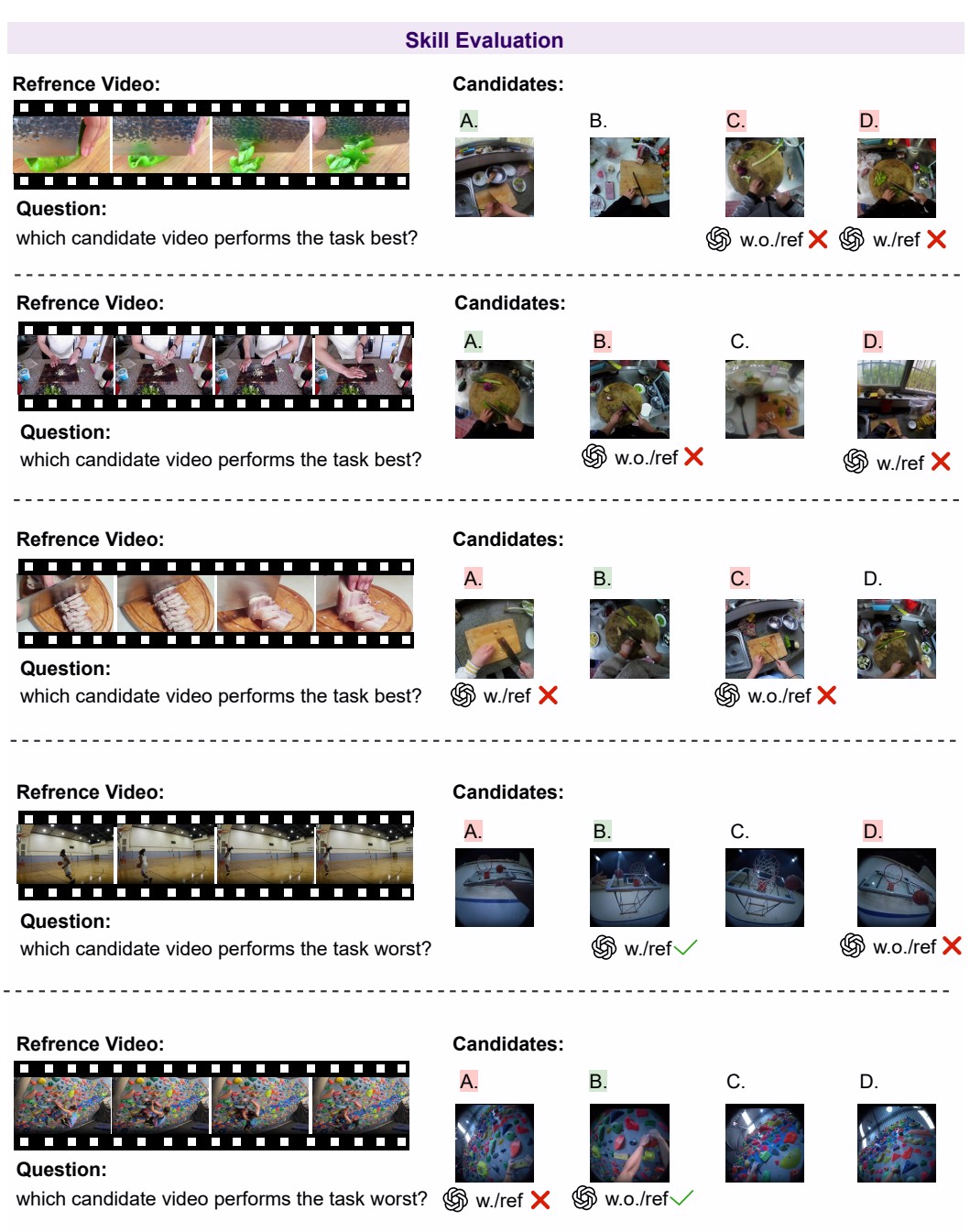

Figure 27: GPT-4o predictions on the Skill Evaluation subtask with and without the reference video. Correct predictions are highlighted in green; incorrect ones in red.

