# OpenReview forum: "EgoExoBench: A Benchmark for First- and Third-person View Video Understanding in MLLMs"
_NeurIPS.cc/2025/Datasets_and_Benchmarks_Track — NeurIPS 2025 Datasets and Benchmarks Track poster_

### Official Review · Reviewer_3E7Q · 2025-06-28

**Rating:** 5
**Confidence:** 3

**Summary:**

EgoExoBench is the first large-scale benchmark focused on evaluating cross-view video understanding in multimodal LLMs. It combines paired egocentric–exocentric videos with over 7,300 multiple-choice questions across eleven subtasks, targeting semantic alignment, spatial reasoning, and temporal integration. Evaluations show that while MLLMs perform well on single-view tasks, they struggle with cross-view reasoning. Techniques like chain-of-thought prompting and cross-perspective cues offer limited improvement, pointing to the need for new architectures that jointly model visual and linguistic information across viewpoints.

**Dataset Code Accessibility:**

No

**Ethical Considerations:**

No, there are no or only very minor ethics concerns

**Final Justification:**

The author's rebuttal has resolved my concerns and I have retained my original rating.

**Limitations Weaknesses:**

- Some omni-modal models capable of processing video, language, and other modalities, such as Qwen2.5-Omni, could also be included in the evaluation to enrich the benchmark's coverage of general-purpose MLLMs.

- The benchmark exclusively adopts multiple-choice question formats, which, while convenient for automatic evaluation, may constrain the models’ expressive potential in real-world cross-view scenarios. It remains unclear whether more open-ended task formats could better reflect the capabilities of MLLMs in such settings.

**Strengths Contributions:**

- This work introduces EgoExoBench, the first benchmark specifically designed for evaluating cross-view understanding in multimodal large language models (MLLMs). It covers 11 fine-grained subtasks across three key dimensions—semantic alignment, spatial correspondence, and temporal reasoning—filling a gap left by existing benchmarks that focus solely on single-view understanding.

- The authors propose a multi-source MCQ construction pipeline that combines annotation-derived, LLM-generated, and human-written questions, supplemented by consistency verification and vision-grounded filtering. This design ensures question diversity and visual dependency while mitigating language-only shortcut biases.

- A comprehensive evaluation of 13 open- and closed-source MLLMs reveals significant performance drops in cross-view tasks compared to single-view ones. This paper further analyzes the limited gains brought by chain-of-thought prompting and additional reference inputs, offering valuable insights for future architecture and training paradigm design.

---

> ### Author Rebuttal · Authors · 2025-07-31
>
> ### **W1**: Some omni-modal models capable of processing video, language, and other modalities, such as Qwen2.5-Omni, could also be included in the evaluation to enrich the benchmark's coverage of general-purpose MLLMs.
>
> Thank you for the constructive suggestion. We now include results for two representative omni-modal models: Qwen2.5-Omni and Ola. Their results are presented in the table below.
>
> To make a meaningful comparison, we also report results for similarly sized vision-language models, Qwen2.5-VL-7B and InternVL3-8B. We find that Qwen2.5-Omni and Ola perform competitively across multiple tasks. In particular, they show stronger performance on Body Part Action Understanding(BPA), Action Ordering(AO), and Sequence Alignment(SA), which require fine-grained visual reasoning and temporal understanding. On other tasks, they achieve performance comparable to other strong baselines.
>
> We agree that omni-modal models are particularly well-suited to the ego-exo benchmark. In real-world applications such as cooking assistance, users may watch third-person instructional videos while wearing egocentric devices that capture their own actions. Intelligent assistants in such settings need to interpret cross-view visual signals, align them with textual or spoken instructions, and provide personalized guidance. This calls for general-purpose MLLMs that handle video, text, and audio seamlessly.
> | Model           | TR | AR | OR | PR | EWI | DP | BPA | AP | AO | SA | SE |
> |----------------|----|----|----|----|-----|----|-----|----|----|----|----|
> | Qwen2.5-Omni    | 30 | 28 | 33 | 27 | 27  | 28 | 40  | 28 | 29 | 47 | 20 |
> | Ola             | 26 | 31 | 37 | 25 | 23  | 29 | 43  | 11 | 29 | 41 | 19 |
> | Qwen2.5VL-7B    | 28 | 27 | 23 | 38 | 24  | 33 | 35  | 20 | 25 | 37 | 24 |
> | InternVL3-8B    | 37 | 32 | 40 | 26 | 20  | 29 | 37  | 31 | 30 | 46 | 23 |
>
>
> We appreciate your suggestion and plan to expand our evaluation to include more omni-modal models in future versions of the benchmark.
>
> ### **W2**: The benchmark exclusively adopts multiple-choice question formats, which, while convenient for automatic evaluation, may constrain the models’ expressive potential in real-world cross-view scenarios. It remains unclear whether more open-ended task formats could better reflect the capabilities of MLLMs in such settings.
>
> Thank you for the constructive comment. Below, we include open-ended tasks experiments and offer our reflections on the trade-offs between closed-ended and open-ended formats for evaluating MLLMs in cross-view video understanding.
>
> **Experiment Setup**:
>
> We randomly sampled 300 questions from three representative tasks in our benchmark: Body Part Action Understanding (BPA), Action Prediction (AP), and Sequence Alignment (SA) (100 questions each). We excluded the other tasks, as their video-matching nature inherently fits the MCQ format and is less suited for open-ended reformulation.
>
> For each question, we removed the answer options and added a brief, task-specific instruction to the prompt to guide the model’s output. For instance, in SA tasks, we asked the model to describe differences in the temporal order of actions between the two videos. These instructions are necessary; without them, models tended to produce vague or irrelevant descriptions. All other settings remained consistent with the original benchmark.
>
> **Evaluation Method**:
>
> To ensure robust evaluation and address potential concerns about automatic scoring, we adopt **a human evaluation protocol**. For each question, two annotators independently compare the model output against the original video content and ground-truth answer. The final score is the average of the two annotators’ ratings based on the following rubric:
>
> - 1 point: The model’s response is mostly correct and aligns with the ground truth.
> - 0.5 points: The response is partially correct but contains inaccuracies.
> - 0 points: The response is incorrect or irrelevant.
>
> We acknowledge that this three-level scoring scheme may not be optimal. We leave more fine-grained evaluations for future work.
>
> **Experimental Results & Analysis**:
>
> The results of four representative models are summarized in the table below:
> | Model                      | BPA     | AP      | SA      |
> |---------------------------|---------|---------|---------|
> | Qwen2.5VL-7B (Closed-set) | 35      | 21      | 37      |
> | Qwen2.5VL-7B (Open-set)   | 30 (-5) | 22 (+1) | 22 (-15)|
> | Qwen2.5VL-72B (Closed-set)| 48      | 46      | 45      |
> | Qwen2.5VL-72B (Open-set)  | 39 (-9) | 40 (-6) | 42 (-3) |
> | InternVL3-8B (Closed-set) | 37      | 31      | 46      |
> | InternVL3-8B (Open-set)   | 29 (-8) | 33 (+2) | 39 (-7) |
> | InternVL3-78B (Closed-set) | 38      | 37      | 52      |
> | InternVL3-78B (Open-set)   | 33 (-5) | 42 (+5) | 40 (-12)|
>
>
> We observe that:
>
> - InternVL3-8B and InternVL3-78B show improved performance in the Action Prediction (AP) task under the open-ended format. This suggests that for tasks involving temporal anticipation, models may benefit from the flexibility to describe expected actions without being constrained by fixed choices.
> - However, in BPA and SA tasks, performance declines across most models. Upon manual inspection, we find that models often describe coarse-level action phases or background differences, rather than the fine-grained action cues required to answer the questions accurately.
>
> **Our Reflections on Open-Ended vs. Closed QA Formats**:
>
> Based on our experimental analysis, open-ended formats can indeed provide a clearer view of the models’ reasoning processes and limitations. However, they often require more careful guidance or task-specific prompting to elicit accurate responses. In contrast, multiple-choice options may help focus the model’s attention by providing structured cues.
>
> Moreover, open-ended tasks introduce additional complexity in evaluation. In this benchmark, we adopt the multiple-choice format as a practical starting point due to its scalability and compatibility with automatic evaluation.
>
> In future work, we plan to explore more refined open-ended task formulations and develop scalable, systematic evaluation method to support them.
>
> We sincerely thank the reviewer again for encouraging us to pursue this important direction.

---

> > ### Author Response · Authors · 2025-08-04
> >
> > Dear Reviewer 3E7Q,
> >
> > Thank you for reviewing our paper. We would greatly appreciate your feedback on whether our rebuttal has addressed your concerns. If any issues remain, we are happy to discuss them further.
> >
> > Thank you for your time and engagement.

---

### Official Review · Reviewer_4K1x · 2025-07-01

**Rating:** 4
**Confidence:** 2

**Summary:**

This paper presents a benchmark to evaluate the MLLMs' ability in understanding and associating exocentric and egocentric videos. The benchmark is built from public ego-exo video datasets, including Ego-Exo4D, EgoExoLearn, LEMMA, EgoMe, TF2023, and CVMHAT. The benchmark is in the form of MCQs, constructed in a hybrid way, including direct derivation, LLM-generation, and human-annotation, adding up to 7300 QA pairs. Multiple MLLMs are evaluated on this benchmark, including both open-source and closed-source ones. GPT-4o-mini leads the leaderboard, and Qwen2.5-VL-72B leads the open-source models. More analysis is provided by investigating the effectiveness of CoT, and reference videos, showing some gaps in current MLLMs development.

**Additional Feedback:**

Although I am familiar with egocentric learning problems, I am no expert in MLLMs. From my understanding of this field, I find the task definitions are not well supported and some of them are not clear. Moreover, authors should try to answer "what makes egocentric viewpoint special/challenging" with experiments and insights, to better motivate and support this work. Therefore, I am giving borderline reject for now.

**Dataset Code Accessibility:**

Yes

**Dataset Code Comments:**

Link to datasets and codes to use it are provided.

**Ethical Comments:**

Only human annotation is involved in this benchmark building. Videos are sourced from public datasets. Therefore, no ethic concerns.

**Ethical Considerations:**

No, there are no or only very minor ethics concerns

**Final Justification:**

I thank authors for their efforts in the rebuttal. I have read it and most of my concerns are addressed. Authors clarify the task definition and provide evidence of the importance of egocentric videos. Given the additional experiments and willingness of revision, I therefore raise the score to borderline accept.

**Limitations Weaknesses:**

I find the major issue with this benchmark is that the proposed "three key dimensions", ego-exo relation, view transition, and temporal reasoning, as well as their corresponding tasks, are not well defined and supported.
- The definition of three dimensions should be revised. The first dimension "ego-exo relation" essentially covers the other two dimensions. The ability to build spatial temporal correspondence is the same as recognizing the same entity or motion across first and third person view.
- For ego-exo relation, the question that is not answered is what makes the egocentric viewpoint special. It can be two exocentric viewpoints videos, evaluating ReID ability. I think authors could add some exocentric QA pairs to explore the difference/ challenge of egocentric viewpoint.
- The same question of "what makes egocentric viewpoint special" can by applied to other tasks, e.g., Body Part Action Understanding, Sequence Alignment, Action Ordering, and Skill Evaluation. Whether it is the egocentric viewpoint making it especially challenging or it is just interleaving video MCQs are challenging is not clear.
- For EWI, could it be the EW wearing a special-looking glasses/equipments that leaks the information?
- Direction prediction should be better explained. I cannot understanding this task by looking at the descriptions and an example in figure 2.
- One final minor point is that I wish to see how much human annotation are involved in this benchmark building. LLMs-generated MCQs can be biased and erroneous.

**Strengths Contributions:**

+ Evaluating the ego-exo video understanding and association is important for the development of contextual AI and spatial intelligence. It has some interesting downstream applications like egocentric skill learning.
+ The efforts in curating this benchmark is appreciated.
+ Analysis in CoT and w/ or w/o reference videos are interesting and offers some insights for the development of egocentric MLLMs.

---

> ### Author Rebuttal · Authors · 2025-07-31
>
> ### **W1**: The definition of three dimensions should be revised.  "ego-exo relation" essentially covers the other two dimensions.
>
> Thank you for this insightful feedback. We agree that clearer definitions are important and appreciate the opportunity to fix and clarify.
>
> **Fix and Clarification of the Three Dimensions**:
> - **Ego-Exo Matching (formerly Ego-Exo Relation)**:
> We rename this to reduce ambiguity and emphasize the core ability to recognize correspondences (objects, actions) between egocentric and exocentric views.
> - **Ego-Exo View Transition**:
> This dimension builds on ego-exo matching, while introducing additional challenge of spatial transformation between the observer (exo view) and the actor (ego view).
>
> - **Ego-Exo Temporal Reasoning**:
> Also building on ego-exo matching, it requires additional reasoning about action temporal order and cross-view action alignment.
>
> **Relationship Among Dimensions**:
>
> Ego-Exo Matching underlies the other two, as both view transition and temporal reasoning depend on correctly associating ego and exo information. In real-world videos, view transition and temporal reasoning are not strictly separate. Thus, we categorize tasks according to the primary type of reasoning they require.
>
> ### **W2**: For ego-exo relation, what makes the egocentric viewpoint special.
>
> Thank you for raising this important point. We agree that clarifying the uniqueness of egocentric viewpoints is essential. Below, we present an exo-exo ablation study and highlight that the challenges and benefits of egocentric views are context-dependent.
>
> **Uniqueness of Egocentric Viewpoints**:
>
> - They capture fine-grained hand-object interactions, which are often missing or less visible in wider third-person shots.
> - Dynamic camera movement causes motion blur and unstable framing.
> - A limited field of view restricts capturing the full body or surrounding context.
>
> **Exo-Exo Ablation Experiment Design**:
>
> To examine the uniqueness of egocentric viewpoints, we conduct an exo-exo ablation study under the ego-exo matching task. In each QA pair, the egocentric video is replaced with a synchronized third-person view. Videos are sourced from LEMMA and EgoExo4D. For a more detailed analysis, we group the videos by scene type: Cooking, Household Chores, Entertainment, Sports, and Assembly. Results are shown in the table below.
>
> | Model               | Cooking | Household | Entertainment | Sport | Assembly    |
> |---------------------|---------|-----------|---------------|-------|-------------|
> | Qwen2.5-VL-7B (ego-exo)   | 47      | 41        | 46            | 27    | 25          |
> | Qwen2.5-VL-7B (exo-exo)   | 49 (+2) | 33 (-8)   | 46 (0)        | 27 (0)| 40 (+15)    |
> | Qwen2.5-VL-72B (ego-exo)  | 61      | 68        | 54            | 30    | 45          |
> | Qwen2.5-VL-72B (exo-exo)  | 58 (-3) | 56 (-12)  | 46 (-8)       | 36 (+6)| 60 (+15)   |
> | InternVL3-14B (ego-exo)   | 47      | 44        | 42            | 26    | 35          |
> | InternVL3-14B (exo-exo)   | 44 (-3) | 41 (-3)   | 50 (+8)       | 34 (+8)| 43 (+8)    |
> | InternVL3-78B (ego-exo)   | 53      | 65        | 59            | 22    | 36          |
> | InternVL3-78B (exo-exo)   | 52 (-1) | 51 (-14)  | 48 (-11)      | 30 (+8)| 48 (+12)   |
> | LLaVA-Video-7B (ego-exo) | 37      | 48        | 30            | 22    | 20          |
> | LLaVA-Video-7B (exo-exo) | 36 (-1) | 33 (-15)  | 26 (-4)       | 28 (+6)| 30 (+10)   |
> | LLaVA-OV-7B (ego-exo)    | 36      | 31        | 35            | 26    | 23          |
> | LLaVA-OV-7B (exo-exo)    | 35 (-1) | 27 (-4)   | 26 (-9)       | 29 (+3)| 33 (+10)   |
>
> **Analysis of Egocentric vs. Exocentric Performance Gaps**:
>
> Egocentric views outperform exocentric ones in cooking, household, and entertainment scenarios, where fine-grained hand-object interactions are critical and better captured from the actor’s perspective.
>
> In contrast, third-person views perform better in sports and assembly tasks, where egocentric videos often suffer from motion blur, occlusions, and a limited field of view that fails to capture full-body movements or the overall workspace.
>
> Based on these observations, egocentric views offer clear advantages in interaction-focused tasks, while exocentric views are better suited for activities requiring stable, wide-angle coverage. To better adapt to real-world scenarios, future MLLMs could adopt viewpoint-aware modeling or fusion strategies to leverage the strengths of both perspectives.
>
>
>
>
> ### **W3**: What makes egocentric viewpoint special for other tasks
>
> Thank you for pointing this out. We also conduct exo-exo ablation studies on: Body Part Action Understanding (BPA), Sequence Alignment (SA), Action Ordering (AO), and Skill Evaluation (SE).
>
> Similar to the ablation study in Ego-Exo Matching, we use synchronized third-person videos to replace the egocentric views in the QA pairs (excluding EgoMe and EgoExoLearn due to the lack of paired exo videos).  Results are shown in the table below.
>
> | Model                | BPA  | AO  | SA  | SE  |
> |----------------------|-----|-----|-----|-----|
> | Qwen2.5-VL-7B (ego-exo)    | 29  | 26  | 37  | 27  |
> | Qwen2.5-VL-7B (exo-exo)    | 37 (+8) | 28 (+2) | 39 (+2) | 32 (+5) |
> | InternVL3-14B (ego-exo)    | 47  | 32  | 47  | 29  |
> | InternVL3-14B (exo-exo)    | 48 (+1) | 52 (+20) | 48 (+1) | 37 (+8) |
> | LLaVA-Video-7B (ego-exo)   | 33  | 28  | 44  | 29  |
> | LLaVA-Video-7B (exo-exo)   | 35 (+2) | 37 (+9) | 45 (+1) | 30 (+1) |
> | Qwen2.5-VL-72B (ego-exo)   | 48  | 34  | 46  | 40  |
> | Qwen2.5-VL-72B (exo-exo)   | 58 (+10) | 48 (+14) | 48 (+2) | 39 (-1) |
> | InternVL3-78B (ego-exo)    | 38  | 32  | 52  | 30  |
> | InternVL3-78B (exo-exo)    | 51 (+13) | 46 (+14) | 54 (+2) | 45 (+15) |
> | LLaVA-OV-7B (ego-exo)      | 25  | 29  | 45  | 27  |
> | LLaVA-OV-7B (exo-exo)      | 25 (0)  | 32 (+3) | 46 (+1) | 27 (0)  |
>
>
> **Analysis**:
>
> In BPA, performance drops when switching from third- to first-person views. This task requires fine-grained left/right body part recognition, which is challenged by egocentric mirror ambiguity and occlusions.
>
> For AO, SA, and SE, most models perform better in exo-exo settings. These tasks involve action localization and sequence comparison. Egocentric videos often suffer from head motion and blurring, disrupting temporal continuity. Third-person views offer stable framing and clearer posture, which benefit event alignment and skill judgment.
>
> **Summary**:
>
> Egocentric views pose challenges like limited field of view, motion blur, and occlusion. The ego-exo performance gap suggests current MLLMs struggle with cross-view understanding. We suggest that enhancing egocentric representation and temporal modeling may help bridge this gap.
>
>
> We will add these analyses in the revision and hope they can address the reviewers' concerns.
>
> ### **W4**: For EWI, could the EW special glasses or equipment leak information?
>
> Thank you for raising this question. We recognize that wearable equipment appears in third-person views, so we have designed safeguards to prevent models from exploiting this shortcut:
>
> - **Multiple wearers**: Each question involves at least two people wearing body cameras, ensuring that equipment alone can’t uniquely identify the ego-wearer.
> - **Hard negatives**: We include other camera-wearing individuals as hard negative options. Experimental and manual checks show models tend to pick the actual recorder rather than relying on equipment appearance.
>
> The table below reports model accuracy, the chance of selecting a hard negative, and the proportion of hard negatives among incorrect answers.
>
> | Model           | Acc (%) | Adv. Rate (%) | Adv. % of Err |
> |-----------------|----------|----------------|----------------|
> | InternVL3-8B    | 20       | 34             | 42             |
> | InternVL3-78B   | 51       | 22             | 45             |
> | LLaVA-OV        | 24       | 30             | 39             |
> | LLaVA-Video     | 29       | 27             | 38             |
> | NVILA           | 26       | 29             | 40             |
> | Qwen2.5VL-7B    | 24       | 32             | 42             |
> | Qwen2.5VL-72B   | 58       | 12             | 28             |
>
>
> **Insights**:
> Weaker models show a higher error rate due to hard negatives. The results suggest that MLLMs often focus on visible, salient objects, which may limit their ability to reason about what lies beyond the frame.
>
> ### **W5**: Direction prediction explanation
>
> The input consists of an egocentric video clip, a question, and four candidate third-person frames. The ego video shows motion of a person or an interacted object. The task is to identify which third-person viewpoint shows the motion in a specified direction (e.g., from left to right).
>
> Figure 2 in the paper illustrates an example: a person walks toward a sink in the ego video. The question asks which third-person view shows the person moving toward the foreground. The four options each show one frame from a different camera with motion directions:
>
>  A) toward background, B) toward left-rear, C) right to left, D) toward foreground
> Thus, the correct answer is D.
>
> We will revise the text and figure for clarity.
>
> ### **W6**: how much human annotation are involved. LLMs-generated MCQs can be biased and erroneous.
>
> Thank you for your question. We clarify that all correct answers in the benchmark (except DP) are derived from ground-truth annotations in the original dataset. For tasks such as TR, AR, OR, PR, EWI, AO, and SE, the distractors are also constructed based on ground-truth annotations provided by the original datasets.
>
> We used human annotators for DP alone, we conducted a five-stage annotation pipeline including Video/Clip selection, QA annotation, question refinement, and QA validation. Two supervisors managed the process and trained three annotators who labeled motion directions, with an additional reviewer verifying the QA quality. The entire process took 120 person hours, 22$/hour is paid as compensation.

---

### Official Review · Reviewer_VqU1 · 2025-07-01

**Rating:** 4
**Confidence:** 5

**Summary:**

This submission introduces **EgoExoBench**, a benchmark that re-packages six public ego–exo video corpora into 7 330 four-choice questions across 11 subtasks grouped under three “cross-view” dimensions (semantic, spatial, temporal), thirteen multimodal LLMs are evaluated zero-shot.

**Additional Feedback:**

* Publish per-question rationales or at least the correct option letter to help users debug.
* Clarify whether option order is shuffled at inference time to avoid positional bias.

**Dataset Code Accessibility:**

Partly

**Dataset Code Comments:**

A light-weight subset and a patched manifest would greatly improve usability.

**Ethical Comments:**

Egocentric clips show faces, homes, and sometimes children, yet no anonymisation procedure or re-consent is described, and IRB approval is absent . The demographic skew toward indoor Western settings (Fig. 3, right) may bake bias into any model tuned on the benchmark. The authors should document participant demographics, apply face blurring where licences allow, and explain compensation and safeguarding of annotators.

**Ethical Considerations:**

Yes, there are ethics concerns that require attention by the authors

**Ethics Flags:**

["Data privacy, copyright, and consent", "Data quality and representativeness", "Discrimination, bias, and fairness"]

**Final Justification:**

I have carefully reviewed all the content from the discussion period. I’m satisfied with the responses to W2, W3, and W4, so I’ve decided to raise my score to 4.

**Limitations Weaknesses:**

1. No new video is collected; the contribution is a layer of MCQs on top of six existing datasets.
2. Distractors and QA filters are generated with Qwen 2.5-32B and DeepSeek-V3 (lines 196-214) , yet the same Qwen family is evaluated (§4.1).
3. Quality control relies mainly on LLM filters; no inter-annotator agreement or error statistics are reported. The human baseline covers only 330 of 7 330 items.
4. MCQs may be solved by option heuristics; the paper gives no adversarial balancing or randomisation analysis.
5. Only zero-shot accuracy is reported; no fine-tuning, retrieval-augmented, or few-shot baselines, and no variance across runs.
6. Prior egocentric–exocentric datasets (e.g., Charades-Ego, Assembly-101) already include paired evaluation tasks, though not in MCQ form.

**Strengths Contributions:**

1. Section 1 persuasively argues that existing video benchmarks treat first- and third-person streams in isolation and therefore cannot test genuine cross-view reasoning.
2. The 11 subtasks (Fig. 2) touch entity matching, wearer ID, spatial mapping, temporal ordering, etc., providing richer coverage than EgoSchema or Video-MME.
3. A three-stage procedure—video collection, QA generation (annotation / LLM / human), and LLM-based filtering—is described with concrete examples and a schematic (Fig. 1).
4. The authors test Chain-of-Thought prompts and remove the reference clip in two subtasks, which gives insight into model failure modes (Table 2).

---

> ### Author Rebuttal · Authors · 2025-07-31
>
> ### **W​1:** No new video is collected; the contribution is a layer of MCQs on top of six existing datasets
>
> We thank the reviewer for the feedback. We first note that constructing benchmarks with no new videos is a widely accepted practice, as seen in EgoSchema, MVBench, EgoTaskQA, EgoMemoria, HourVideo, and many others.
>
> Our contribution is not video collection but the design of a novel benchmark. We address a gap in the video understanding domain: while there are many benchmarks for 1st or 3rd-person videos independently, there is an absence of benchmarks that evaluate cross-perspective understanding between these views.
>
> The VQA format we employ aligns with mainstream video understanding benchmarks, facilitating its integration into current MLLMs evaluation pipelines. Additionally, by unifying six diverse ego-exo datasets under a coherent evaluation protocol, we amplify their impact and enable systematic progress in this underexplored area.
>
> ### **W2:** Distractors and QA filters are generated with Qwen and DeepSeek, yet the same Qwen family is evaluated
>
> **To clarify upfront: Qwen 2.5-32B is not used to generate answers, only for verifying distractor quality and logical consistency. Also, substituting Qwen with other LLMs yields nearly identical results, as we detail below.**
>
> In AO and SA tasks, LLM is used to verify logical consistency. We tested 100 questions across GPT-4o, Claude 3.7, Qwen2.5-72B, and Qwen2.5-32B, and observed 99% consistency. Thus, we chose Qwen2.5-32B for efficiency and cost considerations. In the AP task, all options are drawn from human-annotated actions. LLM is used to help select more challenging distractors. For this, we also found that LLMs achieved 90% consistency, prompting our choice of Qwen2.5-32B.
>
> Additionally, our evaluation includes diverse models beyond the Qwen family to ensure balance. If the reviewer remains concerned about potential bias, we will remove Qwen from the evaluation set.
>
> Finally, we note that this design choice aligns with recent benchmarks (e.g, MVBench, EgoMemoria, TempCompass), where the same LLM family is used for QA filtering and evaluation. That said, we recognize that this is not an ideal setup and would sincerely welcome suggestions for further improving this component of our pipeline.
>
> ### **W3:** Quality control relies mainly on LLM filters; no inter-annotator agreement or error statistics are reported. The human baseline covers only 330 items
>
> We thank the reviewer for the constructive suggestion. We have now substantially expanded the human baseline evaluation from 330 to 2,000 QAs, providing a more comprehensive and statistically meaningful estimate of human performance. The models' relative performance remains unchanged.
>
> QA pairs are derived from existing ground-truth annotations in related datasets. Therefore, the annotation accuracy is guaranteed by construction, and traditional inter-annotator agreement or error rate statistics are not applicable. We will clarify these points in our revised version. Please refer additional explanations in W6 for reviewer 4K1x.
>
> ### **W4 & FB2:** MCQs may be solved by option heuristics; no adversarial balancing or randomisation analysis
>
> Thank you for pointing this out. We address this concern from the following strategies:
>
> **1. Avoiding positional bias during evaluation**
>
> We adopt a Circular Evaluation strategy similar to OpenCompass. Specifically, for each question, we evaluate the model using four permutations of the options: [ABCD, BCDA, CDAB, DABC], and report the average accuracy.
>
> **2. Uniform distribution of correct answers**
>
> To reduce positional bias, correct answers are randomly assigned to one of four positions (A–D), resulting in a near-uniform distribution across all subtasks.
>
> **3. On option heuristics and adversarial balancing**
>
> For tasks like TR, AR, OR, PR, EWI, DP, and SE, all options are videos without text, minimizing reliance on language priors.
>
> In EWI, we include visually misleading distractors based on the presence of the other camera wearer in ego view. The table below reports accuracy, the probability of the model selecting the adversarial distractor, the proportion of adversarial choices among incorrect answers. It shows that lower-performing models are more likely to choose these hard distractors, validating their effectiveness.
>
> | Model           | Acc (%) | Adv. Rate (%) | Adv. % of Err |
> |-----------------|----------|----------------|----------------|
> | InternVL3-8B    | 20       | 34             | 42             |
> | InternVL3-78B   | 51       | 22             | 45             |
> | LLaVA-OV        | 24       | 30             | 39             |
> | LLaVA-Video     | 29       | 27             | 38             |
> | NVILA           | 26       | 29             | 40             |
> | Qwen2.5VL-7B    | 24       | 32             | 42             |
> | Qwen2.5VL-72B   | 58       | 12             | 28             |
>
>
>
> ### **W5:** Only zero-shot accuracy is reported; no fine-tuning, retrieval-augmented, or few-shot baselines, and no variance across runs
>
> Our benchmark is test-only without a training set, making fine-tuning infeasible. To explore task-specific adaptation, we evaluate EgoGPT, an egocentric fine-tuned model. As shown in Table 1 in the table, it performs comparably to similarly sized general models, suggesting limited gains from ego-specific tuning alone. All results reported in the paper are averaged over three runs with different seeds. We will include the standard deviation across runs in the future version.
>
> ### **W6:** Prior egocentric–exocentric datasets (e.g., Charades-Ego, Assembly-101) already include paired evaluation tasks, though not in MCQ form
>
> We appreciate the reviewer’s point and would like to clarify the distinctions:
>
> **Compared with Assembly-101:**
>
> While Assembly-101 includes both egocentric and exocentric videos, its evaluation focuses on single-view action recognition. In contrast, our task explicitly targets cross-view correspondence—requiring the model to identify the same action from a different viewpoint. This directly evaluates a model’s ability to reason across perspectives, not just recognize actions in isolation.
>
> **Compared with Charades-Ego:**
>
> Charades-Ego introduces tasks such as Pairs Discrimination and Moment Localization. However, its moment-level granularity (1-second clips) often lacks sufficient semantic clarity to support complex reasoning. Our action-relation task uses longer, semantically meaningful action clips to enable more reliable evaluation of understanding and reasoning across views.
>
> **Clarifying Motivation and Contribution:**
>
> We agree that prior ego–exo datasets include cross-view tasks. Our key contribution is not the introduction of cross-view evaluation per se, but the design of a standardized, MLLM-compatible benchmark based on the VQA-style multiple-choice format. This structure makes it suitable for evaluating multimodal large language models in a way that existing datasets and tasks do not support directly.
>
> ### **Ethical**
>
> We would like to respectfully correct a misunderstanding in the review. Our benchmark is constructed entirely from existing open-source datasets (e.g., EgoExo-4D, etc.), which were released under research licenses with institutional IRB approval and informed participant consent already in place. We do not collect any new data, nor do we modify the original video content beyond clip selection.
>
> Regarding the claim of demographic skew, we note that the datasets we use span multiple geographic regions, environments (indoor and outdoor), and cultural contexts. In the following table, we list the showing the demographic diversity of our benchmark.
>
> | Activity         | Scene                       | Example                         | #Videos |
> |------------------|-----------------------------|----------------------------------|---------|
> | Household Chores | Living Room, Bedroom        | sweep floor, water plant         | 5735    |
> | Cooking          | Kitchen                     | cook meat, make sandwich         | 8582    |
> | Sports           | Sports field                | soccer, rock climbing            | 3883    |
> | Medical          | Lab                         | peptide synthesis, pipetting     | 1417    |
> | Health           | Indoor room                 | CPR, COVID-test                  | 470     |
> | Assembly         | Workshop, Office            | bike repair, item picking        | 559     |
> | Others           | Playground, Classroom       | take notebook, unlock bike       | 3044    |
>
>
> As for privacy protection, we acknowledge that some clips may contain visible faces or identifiable scenes. Since we are operating within the terms of the original dataset licenses, we follow the anonymization practices provided by those datasets. When redistribution is permitted under license terms, we plan to include face-blurred versions where feasible.
>
> Finally, annotators were compensated at a fair rate of $22 per hour. All annotation work followed standard institutional guidelines to ensure safe and respectful working conditions.
>
> ### **Code**
>
> We would like to correct the reviewer’s impression that the code is only partly available. Our full codebase, including data preparation scripts, evaluation protocols, and model integration code, is available. Moreover, our benchmark has been officially merged into VLMEvalKit, a widely accepted evaluation framework for VLMs. This ensures full open-source, easy to use, and wide compatibility for the community. We will further emphasize this point in the revision.
>
> ### **FB1:** Publish per-question rationales or at least the correct option letter to help users debug
>
> Correct options are already included in our published annotations. Due to word limitations, we’ll provide more detailed explanations and rationales in our revision.
>
> ### **FB2:** Clarify whether option order is shuffled at inference time to avoid positional bias
>
> Yes. Please refer to W5.

---

> > ### Author Response · Authors · 2025-08-04
> >
> > Dear Reviewer VqU1,
> >
> > Thank you for reviewing our paper. We would greatly appreciate your feedback on whether our rebuttal has addressed your concerns. If any issues remain, we are happy to discuss them further.
> >
> > Thank you for your time and engagement.

---

> > > ### Comment · Reviewer_VqU1 · 2025-08-05
> > >
> > > I have carefully reviewed all the content from the discussion period. I’m satisfied with the responses to W2, W3, and W4, so I’ve decided to raise my score to 4.

---

> > > > ### Author Response · Authors · 2025-08-08
> > > >
> > > > Thank you for your careful review and thoughtful comments. We greatly appreciate your feedback and support.

---

### Official Review · Reviewer_ddNe · 2025-07-02

**Rating:** 5
**Confidence:** 4

**Summary:**

This manuscript introduces a large-scale benchmark called EgoExoBench for evaluating cross-first-person and third-person perspective video understanding in multimodal large language models. It primarily evaluates 13 mainstream large models from three aspects: semantic alignment, spatial correspondence, and temporal relation, including 11 subtasks and 7300 Q&As. This manuscript reveals the current limitations of models in cross-perspective reasoning. Despite their excellent performance in single-perspective tasks, these models still haven't reached human-level capability in processing cross-perspective information.

Although EgoExoBench provides comprehensive evaluations of multimodal large language models for cross-view understanding, it would benefit from deeper analysis and a broader range of tasks and datasets to fully realize its potential and broaden its impact.

**Additional Feedback:**

This is a valuable contribution to cross-view video understanding, and EgoExoBench offers a solid benchmark for evaluating MLLMs. However, including more complex real-world scenarios would further strengthen the evaluation. Additionally, a deeper analysis of model architectures and their specific performance on different subtasks could provide valuable insights for future improvements. If these concerns are addressed, I would be happy to increase my rating.

**Dataset Code Accessibility:**

Yes

**Dataset Code Comments:**

The dataset and code for **EgoExoBench** are publicly accessible on Kaggle and Anonymous Github,well-documented, and no known issues related to access

**Ethical Comments:**

There are no significant ethical concerns in this paper as the dataset used is publicly available, and the authors have not raised any issues regarding privacy, consent, or bias in their methodology.

**Ethical Considerations:**

No, there are no or only very minor ethics concerns

**Final Justification:**

The rebuttal addressed my concerns with sufficient depth, especially regarding the role of visual encoding and temporal modeling in model performance. I am increasing my score from borderline accept to accept.

**Limitations Weaknesses:**

- **Lack of in-depth analysis:** While the paper evaluates various models, it lacks an in-depth analysis of model architectures, which could explain performance differences across tasks.
- **Video Background Interference:** Some tasks may be affected by background changes, particularly when distinguishing tasks based on environmental variations (e.g., kitchen scenes).
- **Limited Scope of Tasks and Datasets:** EgoExoBench focuses mainly on standard home and daily activity scenarios, lacking more complex dynamic environments or non-standard settings.

**Strengths Contributions:**

- **Novelty and Significance**: EgoExoBench introduces the first benchmark for cross-view video understanding, addressing three key challenges: semantic alignment, viewpoint association, and temporal reasoning. It holds significant potential to advance embodied agents and human-robot collaboration. The paper also clearly distinguishes itself from single-view video understanding benchmarks (e.g., EgoSchema, Video-MME), emphasizing its unique focus on cross-view video comprehension.
- **Diverse Evaluation**: The manuscript designs 11 question-answer tasks and constructs over 7,300 high-quality multiple-choice questions, ensuring a comprehensive evaluation of multimodal large models’ ability to understand across perspectives.
- **Writing**: The manuscript is well-structured and clear, with effective figures and tables that aid comprehension. The detailed experimental setup and reproducibility guidelines enhance its rigor.

---

> ### Author Rebuttal · Authors · 2025-07-31
>
> ### **W1**: Lack of in-depth analysis
>
> We thank the reviewer for the constructive suggestion. We provide the following analysis and will add them to the revised version to enrich the paper. Below are our observations on the influence of model architectures on different tasks:
>
>  1.  **High-Resolution Visual Encoding is critical**. The ability to process and preserve fine-grained visual detail is a key determinant of success in tasks requiring precise spatial understanding, especially in Ego-Exo View Transition and parts of Ego-Exo Relation. Models like InternVL3 and Qwen2.5-VL, which employ dynamic resolution or tiling strategies to handle high-resolution inputs, consistently outperform models that default to a single, fixed low-resolution input.
> This high-fidelity approach is particularly crucial for:
> - **Egocentric Wearer Identification (EWI) and Body Part Action Understanding (BPA)**, where identifying subtle cues like a partially visible limb or distinguishing between left and right hand actions is paramount.
> - **Object Relation (OR) and Person Relation (PR)**, which demand the matching of specific instance-level features (e.g., clothing textures, object shapes) across different viewpoints.
> Conversely, models like LLaVA-OV, which aggressively compress visual tokens via interpolation for computational efficiency, demonstrate weaker performance on these tasks. This suggests that while such compression is viable for general scene understanding, it creates an information bottleneck that discards the critical details necessary for fine-grained cross-view association.
> 2. **Temporal Modeling is useful**.
> Performance in the Ego-Exo Temporal Reasoning category is strongly correlated with the sophistication of the model's temporal encoding mechanism. Qwen2.5-VL's architecture introduces an absolute temporal encoding strategy. This provides the model with a physical sense of temporal flow and duration, a distinct advantage over simpler sequential encoding that merely counts frames. This architectural choice likely explains Qwen2.5-VL's superior performance on:
> - **Action Ordering (AO)**, where understanding the relative timing and pace of actions across asynchronous or differently edited videos is essential.
> - **Action Prediction (AP)**, as forecasting the next step benefits from an understanding of the current action's temporal rhythm.
> In contrast, architectures that employ simplistic temporal modeling, such as uniform sequential encoding or, more drastically, temporal-dimension average pooling (e.g., NViLA), effectively erase nuanced temporal dynamics. Such methods struggle to disambiguate the order of events, leading to significantly lower accuracy on all temporal reasoning subtasks.
>
> ### **W2**: Video Background Interference
>
> We thank the reviewer for raising this important point. We acknowledge that background variation can be a confounding factor, and we have carefully designed our benchmark to minimize the impact of background variation. Specifically, all questions and candidate videos are drawn from the same scene types, ensuring that models cannot rely on superficial background cues. For distinguishing tasks, we design specific and stricter controls for each task, which are summarized in the table below.
>
> Definitions upfront for clarity:
> - Scene type: high-level semantic scenario categories such as "kitchen" or "lab."
> - Environment: specific physical locations (e.g., kitchen 1 vs. kitchen 2).
> - Same environment , different camera viewpoints: videos captured in the same physical locations (e.g., one lab) using different viewpoints.
> - Same video: the options are drawn from the same original video file.
>
> | SubTask Type                              | Source Setting                         | Scene Consistency Strategy                                                                 | Remarks                                                                                     |
> |-------------------------------------------|----------------------------------------|---------------------------------------------------------------------------------------------|---------------------------------------------------------------------------------------------|
> | Object Relation, Action Relation (LEMMA)  | Same video                             | No background variation                                                                     | All candidates from same video; GT-candidate from temporally-aligned clip; negatives differ temporally |
> | Person Relation, Egocentric Wearer ID     | Same video                             | No background variation                                                                     | All candidates from same video and same clip; differ only by bounding boxes                |
> | Action Relation (EgoExoLearn)             | Same scene type, different environment | All candidates from same scene type (e.g., kitchens or labs), but different locations       | Tests generalization of action understanding across environments                            |
> | Direction Prediction, Action Ordering     | Same environment, different viewpoints | No background variation                                                                     | All videos from same physical environment; vary only by viewpoint (e.g., ego vs. exo)       |
> | Task Relation                             | Same scene type, different environments| All videos from the same scene type                                                         | Models must rely on high-level task semantics, not background                               |
> | Sequence Alignment, Action Prediction, Skill Evaluation | Same scene type               | All videos from the same scene type                                                         | Evaluates ability to align or predict actions/skills across similar environments            |
>
> In summary, background interference is minimized through careful control over scene types and environments, ensuring that each task evaluates the intended reasoning skill rather than superficial visual cues.
> We will provide a clearer description of these design choices in the revised version.
>
> ### **W3**: Limited Scope of Tasks and Datasets
>
> Thank you for raising this important concern. We agree that data diversity is crucial for robust evaluation. We would like to clarify that we have made efforts to ensure diverse and realistic coverage. We integrate videos from six public egocentric-exocentric datasets. All videos are from real-world scenarios.
>
> We appreciate your suggestion and will add data statistics in the future version.
> - **Scene Diversity**: Our dataset spans indoor and outdoor environments, including homes, labs, workshops, and sports fields. These go well beyond home scenarios.
> - **Activity Diversity**: The benchmark covers not only daily activities but also sports (e.g., basketball, football, climbing), medical and laboratory procedures (e.g., COVID testing, scientific experiments), and assembly tasks (e.g., bicycle repair), which go beyond typical household activities.
> - **Egocentric Viewpoint Challenges**: Egocentric videos naturally involve issues such as camera shake, motion blur, partial occlusion, and limited field of view, especially in complex dynamic environments. For example, in the climbing scenes, hands or feet may move outside the camera’s field, adding difficulty in recognizing complete actions.
> To further clarify the scope of scenarios covered, we also provide a table that categorizes the benchmark videos by scene and activity type.
> | Activity         | Scene                                | Example                                | # Videos |
> |------------------|--------------------------------------|--------------------------------------- |----------|
> | Household Chores | Living Room, Bedroom, Dormitory, etc | sweep floor, water plant, change water | 5735     |
> | Cooking          | Kitchen                              | cook meat, make sandwich, make noodle  | 8582     |
> | Sports           | Sports field (indoor & outdoor)      | soccer, basketball, rock climbing      | 3883     |
> | Medical          | Lab                                  | solid-phase peptide synthesis          | 1417     |
> | Health           | Indoor room                          | CPR, COVID-test                        | 470      |
> | Assembly         | Workshop, Office                     | bike repair, pick items into bags      | 559      |
> | Others           | Playground, Library, Classroom, etc  | take notebook, unlock a shared bike    | 3044     |
>
> To the best of our knowledge, this represents one of the most diverse and comprehensive collections currently available for evaluating egocentric-exocentric reasoning in MLLMs. We are aware that this setup is not yet optimal, and would greatly appreciate any suggestions for additional datasets or tasks to incorporate in future versions.

---

> > ### Comment · Reviewer_ddNe · 2025-08-07
> >
> > The authors' rebuttal has effectively addressed some of my concerns. The analysis of how different model designs for visual encoding and temporal modeling impact performance is now more thorough. I have therefore decided to raise my score. I believe incorporating this part into the revised version of the paper will further strengthen its contribution and provide valuable insights for future research.

---

> > > ### Author Response · Authors · 2025-08-08
> > >
> > > Thank you for your constructive feedback and helpful suggestions. We will incorporate this analysis into future revisions of the paper.

---

> ### Author Response · Authors · 2025-08-04
>
> Dear Reviewer ddNe,
>
> Thank you for reviewing our paper. We would greatly appreciate your feedback on whether our rebuttal has addressed your concerns. If any issues remain, we are happy to discuss them further.
>
> Thank you for your time and engagement.

---

### Note · Authors · 2025-08-12

Dear Area Chairs and Reviewers,

We sincerely thank the reviewers and the Area Chair for their constructive feedback and valuable suggestions, and we have carefully addressed the concerns in our rebuttal.

First, we summarise the key strengths of our work as recognised by the reviewers:

1. **Novelty**: EgoExoBench is the first benchmark specifically designed to evaluate cross-perspective video understanding in MLLMs, setting it apart from single-view benchmarks.

2. **Breadth**: It covers three core ego–exo tasks and eleven subtasks, spanning diverse scenarios and activities.

3. **Insight**: Our analyses of Chain-of-Thought prompting and cross-perspective guidance offer valuable guidance for future research.

Then, we also address the core concerns raised by the reviewers with targeted clarifications and supplementary evidence.

1. **Data Quality Control**
  - Clarified strategies to reduce background variation interference. (**For ddNe**)
  - Confirmed all correct answers are human-annotated. (**For VqU1, 4K1x**)
  - Clarified hard negative strategies in the EWI task to remove potential shortcuts. (**For 4K1x**)
2. **Deeper Analysis**
  - Analyzed the influence of different model architectures on task performance. (**For ddNe**)
  - Conducted exo–exo ablation to highlight the unique challenges of egocentric viewpoints. (**For 4K1x**)
3. **Evaluation Protocol**
  - Clarified MCQ bias reduction strategies, including circular evaluation, uniform correct answer distribution, and video-only option design for most tasks (TR, AR, OR, PR, EWI, DP, SE). (**For VqU1**)
  - Conducted open-ended task experiments and discussed trade-offs between open-ended and MCQ formats, showing that MCQs remain an efficient choice for cross-view evaluation. (**For 3E7Q**)

We appreciate that most reviewers have provided positive feedback after considering our rebuttal. In our rebuttal, we have made detailed clarifications and conducted additional analyses and experiments to thoroughly address the reviewers’ concerns, especially regarding data quality and the evaluation protocol. We hope that our efforts will be helpful for the final discussion and consideration of our work.

---

### Decision · Program_Chairs · 2025-09-18

**Decision:**

Accept (poster)

**Comment:**

This paper receives an overall rating of 5/5/4/4/4. It proposed a novel benchmark to facilitate the understanding of videos under different (first-person and third-person) viewpoints -- this is a solid contribution to the community. The AC reads the paper and discussions, and agrees with the unanimity of reviewers to accept the paper for publication.